# Nuclear ingression of cytoplasmic bodies accompanies a boost in autophagy

Manon Garcia[1],*, Sylvain Kumanski[1],*, Alberto Elías-Villalobos[2,3] ⓘ, Chantal Cazevieille[4], Caroline Soulet[1], María Moriel-Carretero[1] ⓘ

Membrane contact sites are functional nodes at which organelles reorganize metabolic pathways and adapt to changing cues. In *Saccharomyces cerevisiae*, the nuclear envelope subdomain surrounding the nucleolus, very plastic and prone to expansion, can establish contacts with the vacuole and be remodeled in response to various metabolic stresses. While using genotoxins with unrelated purposes, we serendipitously discovered a fully new remodeling event at this nuclear subdomain: the nuclear envelope partitions into its regular contact with the vacuole and a dramatic internalization within the nucleus. This leads to the nuclear engulfment of a globular, cytoplasmic portion. In spite of how we discovered it, the phenomenon is likely DNA damage-independent. We define lipids supporting negative curvature, such as phosphatidic acid and sterols, as bona fide drivers of this event. Mechanistically, we suggest that the engulfment of the cytoplasm triggers a suction phenomenon that enhances the docking of proton pump-containing vesicles with the vacuolar membrane, which we show matches a boost in autophagy. Thus, our findings unveil an unprecedented remodeling of the nucleolus-surrounding membranes with impact on metabolic adaptation.

## Introduction

Eukaryotic cells possess a functionally committed system of endomembranes whose regulated remodeling is essential to warrant adaptation to stresses, changing cues and cell cycle requirements. Among them, the ER is one of the most dynamic, suffering drastic transitions during ER stress, when the volume of membranes massively expands to increase its protein folding capacity (Ron & Walter, 2007). The perinuclear subdomain of the ER, also known as the nuclear envelope, is particularly prone to extreme remodeling. Irrespective of nuclear division occurring in an "open" or in a "closed" manner, the perinuclear ER membranes will undergo dramatic changes either because of rupture, dispersion, and re-assembly or because of expansion and deformation (Zhang & Oliferenko, 2013; Ungricht & Kutay, 2017). More specifically, in *Saccharomyces cerevisiae*, the subdomain of the nuclear envelope surrounding the nucleolus can experience massive membrane expansion (Campbell et al, 2006; Witkin et al, 2012) and is the key for nuclear organization in response to the metabolic status (Barbosa et al, 2019).

Part of the process of membrane remodeling is governed by the physical proximity between different endomembrane systems, known as membrane contact sites (Scorrano et al, 2019). At these locations, membranes belonging to at least two different organelles, such as the ER and the Golgi, or the mitochondria and the lipid droplets, stay in close proximity (10–80 nm), which allows the spatial organization of enzymes involved in a given metabolic pathway (Rogers et al, 2021) and the active exchange of lipids and ions (Lahiri et al, 2015). In the case of the nucleolus-surrounding nuclear envelope subdomain, its contact with the membrane of the vacuole (the equivalent in *S. cerevisiae* to the lysosome), termed the nucleus–vacuole junction (NVJ), is of particular importance. The tether between these two membranes is actively maintained by the proteins Vac8 and Nvj1, present in the vacuolar and outer nuclear membranes, respectively (Pan et al, 2000). Helped by additional factors, such as Snd3 (Tosal-Castano et al, 2021), the NVJ expands "zipper-wise" during nutritional stress, such as glucose and amino acid shortage, and upon target of rapamycin complex (TORC) inhibition (Hariri et al, 2018). This increase in the contact surface can serve to send esterified lipids for storage within lipid droplets (Hariri et al, 2018), to increase the flux of metabolites through the mevalonate pathway (Rogers et al, 2021), to modulate the sphingolipid biosynthetic pathway (Kvam & Goldfarb, 2006; Henne et al, 2015), or to recycle nonessential nuclear components (Kvam & Goldfarb, 2007), such as nucleolar proteins and forming ribosomes, as to decrease the cell's translation capacity (Kvam & Goldfarb, 2007; Mostofa et al, 2018; Mostofa et al, 2019). Furthermore, the NVJ is important to alleviate stress associated to defective nuclear pore complex assembly (Lord & Wente, 2020).

[1]Centre de Recherche en Biologie cellulaire de Montpellier (CRBM), Université de Montpellier, Centre National de la Recherche Scientifique, Montpellier, France   [2]Instituto de Biomedicina de Sevilla (IBiS), Hospital Virgen del Rocío-CSIC-Universidad de Sevilla, Sevilla, Spain   [3]Departamento de Genética, Universidad de Sevilla, Sevilla, Spain   [4]Institut de Neurosciences de Montpellier (INM), Université de Montpellier, INSERM, Montpellier, France

Correspondence: maria.moriel@crbm.cnrs.fr
*Manon Garcia and Sylvain Kumanski contributed equally to this work.

In this work, we define a new remodeling event at the nucleolus-surrounding nuclear envelope by which this subdomain partitions between its contact with the vacuole and a drastic internalization within the nucleus. We find that enrichment at membranes in phosphatidic acid or free sterols, which support negative curvature, promotes this phenomenon. This leads to the nuclear engulfment of a globular cytoplasmic portion at a very short distance from the vacuole. Presumably by a suction phenomenon, this fosters the docking of proton pump-containing vesicles with the vacuolar membrane, which matches a boost in the potential to accomplish general autophagy. We thus unveil an unforeseen membrane-remodeling event with an impact on metabolic adaptation and raise the question of how the invasion of the nucleoplasmic space by such voluminous bodies affects genome homeostasis.

## Results

### Identification of unusual structures inside the nucleus during treatment with different genotoxic agents

To undoubtfully identify the nucleus when monitoring the formation of foci by DNA repair factors, we routinely use the nucleoplasmic protein Pus1 tagged with mCherry at its N-terminal region (Fig S1A). We realized that, when some genotoxic agents are added to cell cultures growing exponentially in a rich medium, the Pus1 signal in some nuclei became displaced, or even absent, giving rise to what we informally defined as "holes" (Figs 1A and S1B). This serendipitous but repetitive observation prompted us to investigate the nature of such structures. We therefore systematically

quantified the percentage of cells displaying at least one of these holes upon exposure to different genotoxic agents. We found that, already basally, 6% of the cells manifested this phenomenon. Methyl methanesulfonate (MMS) modestly doubled this percentage, and most pronouncedly, zeocin triggered a time-dependent increase in the frequency of these structures (Fig 1B). On the contrary, 4-nitroquinoline 1-oxide (4-NQO), camptothecin (CPT), and hydroxyurea (HU), genotoxins that affect DNA differently, did not induce the formation of these structures (Fig 1B). All the genotoxic agents used in this set-up provoke cells arrest in different stages of the cell cycle, which validated their activity (Fig 1C, red asterisks). These data suggest that the "nuclear holes" phenotype could be unrelated to DNA damage. Zeocin has been reported to provoke an arrest in $G_2$ during which DNA segregation toward the daughter cell is paused, whereas nuclear envelope expansion continues, leading to the accumulation of overgrown nuclear membranes, a phenotype that is not triggered by HU (Witkin et al, 2012). Yet, although cell arrest in $G_2$ could be a necessary trigger, it is manifestly not sufficient, for cells treated with CPT or 4-NQO also accumulate in $G_2/M$ (Fig 1C) without displaying "nuclear holes" (Figs 1B and S1B).

### Phosphatidic acid supports the formation of zeocin-triggered nuclear holes

Given the strength of the phenotype, we focused on the treatment with zeocin to further characterize the phenomenon of nuclear hole formation. As cell cycle progression relates to nutrient availability, we first tested whether nutritional conditions could have any impact on the apparition of these structures. Importantly, in

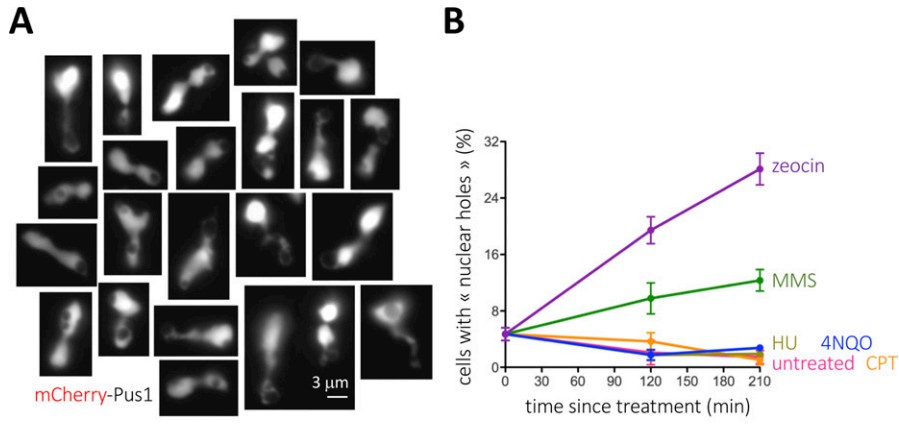

**Figure 1. Detection of nuclear holes in response to genotoxic agents.**
**(A)** Illustrative images of mCherry-Pus1 signals (nuclei) from *Saccharomyces cerevisiae* cells exposed to 100 µg/ml zeocin for 210 min in which "black holes" can be observed. Eventual saturated images are so to permit the delineation of the Pus1 signal surrounding the holes. **(B)** Quantification of the percentage of cells displaying nuclear holes in response to the indicated genotoxic agents at the indicated time-points. The used doses were 100 µg/ml zeocin, 100 mM HU, 0.1% MMS, 100 µM CPT, and 0.05 mg/l 4-NQO. The plotted values and the error bars are the mean and the SEM, respectively, of three independent experiments. At least 200 cells were counted per time-point, treatment and experiment. **(B, C)** Cytometry profiles of the experiment shown in (B). 1c and 2c indicate the DNA content. Red asterisks mark the time-points when alterations in the cell cycle profiles can be detected, as compared with the untreated samples.

marked contrast to what we observed in a rich medium, growing the cells in a defined, minimal medium abolished the formation of nuclear holes in response to zeocin (Fig 2A, pEmpty; Fig S2A). Of note, zeocin was efficiently incorporated in minimal medium-growing cells, as demonstrated by its ability to provoke a $G_2$ cell arrest and to engage the DNA damage sensing factor Tel1 into forming zeocin-induced foci (Fig S1C). Together, this suggests that a factor(s) needed to form the holes, presumably abundant in a rich medium, may become limiting in a minimal medium. Although the origin of the holes could be diverse, it was likely that they derive from transitions at the nuclear envelope. Lipids are among the many molecules needed to remodel membranes whose availability is impacted by nutritional conditions. In that context, lipids supporting negative curvature (Fig 2B, left, yellow phospholipids), such as phosphatidic acid (PA) or diacylglycerol (DAG) (Ben M'barek et al, 2017; Choudhary et al, 2018), are expected to be relevant at specific sites, both at the outer and the inner nuclear membranes (ONM, INM), to trigger invagination (Fig 2B, right, yellow phospholipids). Overexpression for 2 h of the PA-generating enzyme Dgk1 (Fig 2C) led to an imperceptible increase in the presence of nuclear holes (Fig 2A, p$DGK1^{OE}$, time "0"). Yet, maintaining the overexpression 210 min longer led to 35% of cells displaying nuclear holes (Fig 2A, p$DGK1^{OE}$, time "210*"). Moreover, addition of zeocin to Dgk1-overexpressing cells doubled the percentage of cells displaying nuclear holes in only 20 min despite growth occurring in a minimal medium and led to a final 55% of cells bearing the phenotype (Fig 2A, p$DGK1^{OE}$, times "20-210"; Fig S2A). On the contrary, overexpression of the hyperactive Pah1 allele Pah1-7A, which promotes the accumulation of DAG at the expense of PA (Fig 2C), did not provoke any increase in the percentage of cells displaying nuclear holes, even after 210 min since zeocin addition (Fig 2A, p$PAH1$-$7A^{OE}$; Fig S2A). These data point at PA as relevant to form nuclear holes and exclude DAG molecules as implicated in this process, in spite of their conical shape promoting negative curvature (Choudhary et al, 2018). In support, the reciprocal approach using deletion mutants demonstrated that an excess of PA creates a constitutive presence of nuclear holes in approximately half of the population (Fig 2D, pah1Δ), but the chronic excess of DAG only slightly yet not significantly differed from the isogenic WT strain (Fig 2D, dgk1Δ). Alternatively, the role of PA and the lack of effect of DAG suggest that the proneness to form the "holes" could relate to an excess of available phospholipids.

If PA is important to trigger transitions at the nuclear envelope in response to zeocin, then zeocin treatment is expected to trigger PA accumulation or redistribution. We used two versions of a fluorescent PA biosensor (description in the legend of Figs 2E and S2B, [Romanauska & Köhler, 2018]). When expressed to access the full WT cell, this sensor mainly binds the plasma membrane and yields a faint nucleoplasmic signal, as described (Loewen et al, 2004). When fused to a NLS, it provides a basally nucleoplasmic diffuse signal (Fig 2E, top left; Fig S2B) (Romanauska & Köhler, 2018). Its performance/specificity can be controlled by raising PA levels (either through Dgk1$^{OE}$ or Pah1 absence), which thus leads to a very strong deformation of the nuclear contour and to the binding of the sensor to the nuclear envelope (Fig S2B). We treated cells growing exponentially in a rich medium with zeocin and monitored the localization of both PA biosensor signals in time. The addition of zeocin led to the clustering of the cell-wide biosensor either all over the nuclear envelope or at discrete spots distributed all around the nuclear periphery, suggesting concentration or at least exposure of PA at those sites (Fig S2C). Importantly, the nucleus-targeted biosensor signals became perinuclear (Fig 2E), peaking at 60–100 min after zeocin treatment (Fig 2F). Later, the PA biosensor progressively became nucleoplasmic again, indicative of PA detection at the INM being transient (Fig 2F). As a control for the specificity of this behavior, we were unable to observe this transient signal enrichment at the INM if the experiment was performed in a minimal medium (Fig S2D). Thus, PA seems to be a key molecule in promoting the formation of nuclear holes, in general, and in response to zeocin, in particular.

## Sterols cooperate in the formation and maintenance of nuclear holes

Apart from PA or DAG, sterols promote negative curvature at biological membranes (Ben M'barek et al, 2017), and in agreement, we have recently shown that they can support the internalization of nuclear envelope-derived nuclear lipid droplets (Kumanski et al, 2022). To increase the concentration of sterols in membranes and assess their putative contribution to nuclear holes formation, we used mutant strains either impaired in the storage within cytoplasmic LD of free sterols (are1Δ are2Δ, simplified in the literature as steΔ) (Hongyuan et al, 1996) or unable to release sterols from membranes (yeh2Δ) (Müllner et al, 2005). Importantly, both mutants modestly but reproducibly displayed nuclear holes in a basal manner (Fig 2G). Addition of zeocin led to an additive formation of nuclear holes (Figs 2G and S2E). Thus, the free sterols embedded in membranes are not needed to form nuclear holes in response to zeocin yet can act as adjuvants or even as independent elicitors, perhaps by favoring a membrane context that is more susceptible to their genesis.

## Kinetics and nature of nuclear hole formation

To gain some insights into the process of nuclear hole formation, we monitored by total interference reflection fluorescence (TIRF) time-lapse microscopy their apparition, thanks to the GFP-tagging of the nucleoplasmic protein Nab2, while defining the nuclear periphery with a fluorescently tagged nucleoporin, Nup57-tDIMER (Wang et al, 2016). By using yeh2Δ cells exposed to zeocin, we found that the holes are bodies that internalize from the cytoplasm in a process that takes from 15 to 30 min (Fig 3A). Observation of the tagged nucleoporin revealed that this can happen either after nucleoporins split apart from the site of ingression (Fig 3A, examples "1" & "2," arrowheads) or by transient invagination of the nuclear membranes that are still decorated with nucleoporins (Fig 3A, examples "3" & "4," arrowheads). These latter events suggest that the nuclear holes are delimited, at least initially, by the nuclear envelope. In this sense, through-focus serial acquisitions of Nab2-GFP signals demonstrated that the holes are almost perfectly spherical intrusions from the cytoplasm that nevertheless remain connected to it by thin, rope-like bridges (Fig 3B, arrowheads). Last, electron microscopy confirmed not only the presence within the nucleus of bodies of electron density similar to that of the

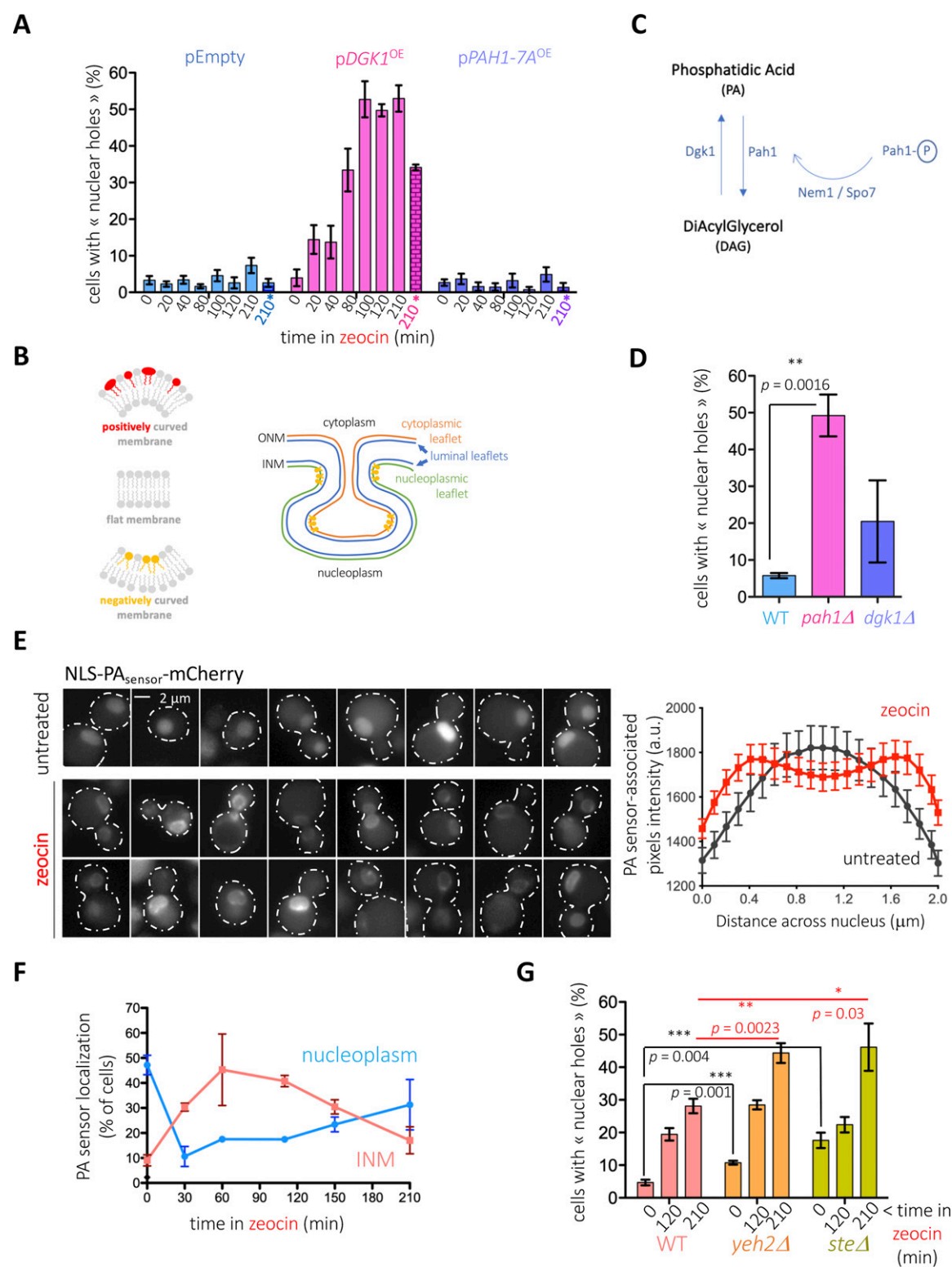

**Figure 2. Lipid determinants of nuclear holes formation.**
**(A)** Cells bearing the genomic mCherry-Pus1 construct were grown overnight in a minimal medium selective for the indicated plasmids with glycerol as the carbon source. The exponential cultures were then supplemented with 2% galactose to induce the expression of nothing (pEmpty), of Dgk1 (pDGK1$^{OE}$), or of the hyperactive Pah1-7A (pPAH1-7A$^{OE}$). Two hours later (time "0"), 100 μg/ml zeocin was added. The indicated time-points therefore indicate the elapsed time since zeocin addition, except for the last point of each set ("210*" and marked in color), which accounts for the impact of the overexpression only. The percentage of cells in the population displaying at least one nuclear hole was counted. Each bar reflects the mean of three independent experiments, and the error bars account for the SEM. At least 200 cells were considered per time point, condition and experiment. **(B)** Left: simplified scheme of basic membrane curvature set-ups in which conical phospholipids (in yellow) help

cytoplasm (Fig 3C) but also that such bodies are delimited by two membranes, again pointing at the nuclear envelope as the boundary of the holes (Fig 3D).

### Nuclear holes preferentially initiate from the nucleolus-wrapping envelope subdomain

The nuclear membranes associated to the nucleolus, which hosts the rDNA, is repeatedly reported both as prone to expansion and to support lipid transitions (Campbell et al, 2006; Witkin et al, 2012; Barbosa et al, 2019; Walters et al, 2019). We therefore assessed the formation of nuclear holes with respect to this nuclear envelope subdomain by relating mCherry-Pus1–defined holes to the Net1-GFP–marked rDNA position (Matos-Perdomo & Machín, 2018). Time-lapse microscopy studies revealed that the ingression of the holes systematically occurs in close association to the Net1-GFP marked rDNA and even led to perturbations in the original shape of Net1 patterns (Fig 4A). These data suggest that even if PA concentration is detected all over the INM (Fig 2E and F), it is more likely to give rise to invagination events at the subdomain surrounding the nucleolus.

The $G_2$-associated expansion of the nucleolus-enveloping membranes is also elicited by the microtubule-polymerizing inhibitor nocodazole (Witkin et al, 2012). Furthermore, this nocodazole-triggered membrane expansion has been recently discovered to instruct the formation of structures reminiscent to nuclear holes in a process during which the protrusion circumvents and wraps the vacuole, using it as a template (Matos-Perdomo et al, 2021 Preprint). We treated cells for 210 min with nocodazole, which was enough to accumulate cells in $G_2/M$ (cytometry profiles, Fig 4B) and found a very robust induction of this phenotype (Fig 4B). In agreement with this recent report (Matos-Perdomo et al, 2021 Preprint), we undoubtfully detected by electron microscopy tongue-like nuclear protrusions curling around the vacuole (Fig 4C), meaning that, even if both events stem from the nuclear envelope subdomain adjacent to the nucleolus, zeocin- and nocodazole-induced phenomena are of a different nature.

We last assessed the relative position of holes at a given time to establish whether, once inside, the holes remain within the nucleolus. To do so, we used two fluorescent nucleolar markers: either Nop1-CFP, a protein soluble in the nucleolus (Mekhail et al, 2008), or again Net1-GFP, the rDNA-bound factor. We found that, irrespective of the marker, the nucleolus was frequently close to the holes (Fig 4D), yet the percentage of cells displaying the black hole inside the nucleolus (i.e., the hole was irrefutably in the middle of the nucleolar signal) accounted for a small percentage of all the events, and this even in genetic contexts where black holes were very frequent, such as in the pah1Δ strain, or in its genetic mimic nem1Δ (Fig 4E). Thus, these data suggest that the holes display a mobility that makes them diffuse away from the nucleolar, entry-point subdomain.

### Links between nuclear holes and vacuoles

Nuclear envelope protrusions leading to a hole display a link with vacuoles (Matos-Perdomo et al [2021] Preprint and Fig 4B and C), and the nuclear envelope subdomain surrounding the nucleolus, which we define as prone to cytoplasmic ingression (Fig 4A), is also involved in NVJ establishment (Pan et al, 2000). As such, we hypothesized that the NVJ status may matter during nuclear-hole formation. To assess this, we disrupted the establishment of the NVJ by deleting NVJ1 and counted the percentage of cells capable of forming holes in response to zeocin. Interestingly, cells formed nuclear holes more readily in the absence of Nvj1 (Fig 5A, from 0 to 120 min treatment in WT [twofold] versus nvj1Δ [11-fold]), although the total final number was unchanged (Fig 5A, time 210 min). In a yeh2Δ strain, where the basal level of holes and the readiness to form them upon zeocin treatment is increased, the absence of Nvj1 did not strengthen the phenotype further (Fig 5A, yeh2Δ versus yeh2Δ nvj1Δ). Together, these data suggest that the nuclear envelope subdomain that surrounds the nucleolus is shared between its contact with the vacuole and its availability to engage in nuclear-hole formation.

To further explore the physical relationship of the nuclear holes and the vacuoles, we re-inspected our images and observed that some black holes seemed to correspond to vacuoles in the DIC channel (Fig 5B). These events neatly differed from situations where the vacuole is so big that it pushes, therefore deforms, the nucleus (Fig S3A). Yet, our electron microscopy data argued against zeocin-

---

shape membranes of negative curvature, cylindrical phospholipids (in gray) give rise to flat membranes, and inverted conical phospholipids (in red) serve to shape positively curved membranes. Right: invaginations (for simplicity only of the inner nuclear membrane, INM) request negative curvature-promoting phospholipids (in yellow). This requirement is modest at regions such as the luminal leaflet of the INM, whereas it is maximal at nascent sites in the nucleoplasmic leaflet. **(C)** Simplified scheme illustrating the enzymes responsible for PA and DAG synthesis. Pah1 is subjected to inactivation by phosphorylation (Pah1-P). To bypass the need of the phosphatase Nem1/Spo7 complex to activate it, we have overexpressed a constitutively dephosphorylated isoform, Pah1-7A. **(D)** Cells of the indicated genotypes bearing the genomic mCherry-Pus1 construct and growing exponentially in the rich medium were photographed and the percentage of cells in the population displaying at least one nuclear hole was counted. Each bar reflects the mean of three independent experiments, and the error bars account for the SEM. At least 200 cells were considered per condition and experiment. The P-value indicates the statistical significance upon performing a t test. **(E)** (Left) Exponentially growing (in rich medium) WT S. cerevisiae cells transformed with a previously validated, nucleus-directed, mCherry-tagged sensor capable of detecting membrane-bound phosphatidic acid (NLS-PA$_{sensor}$-mCherry [Romanauska & Köhler, 2018], derived from the Q2 domain of Opi1 [Loewen et al, 2004]) were treated (or not) with 100 μg/ml zeocin and inspected by fluorescence microscopy. Representative raw images are displayed. Please note that different intensities among cells may be because of the biosensor being expressed from a plasmid. (Right): a line was drawn through nuclei using the raw images, and pixel intensity values across the line (distance) were plotted for both zeocin (red line) and untreated (black line) conditions. The graph displays the mean intensity and the SEM for n = 27 for untreated, n = 44 for zeocin. **(E, F)** The same WT cells illustrated in (E) were followed in time after zeocin addition. The percentages of cells displaying either nucleoplasmic (blue line) or perinuclear (INM: inner nuclear membrane, pink line) localization of the PA-associated signal are plotted. Please note that the addition of both nucleoplasmic and perinuclear percentages does not reach 100%. This is because of the presence in the population of cells displaying either lack of signal or vacuolar signal (presumably because of sensor degradation). The plotted values are the mean and the SEM of three independent experiments. **(G)** Cells of the indicated genotypes bearing the genomic mCherry-Pus1 construct and growing exponentially in the rich medium were photographed before 100 μg/ml zeocin addition and 120 and 210 min later. The percentage of cells in the population displaying at least one nuclear hole was counted. Each bar reflects the mean of four to seven independent experiments, and the error bars account for the SEM. At least 200 cells were considered per condition and experiment. The P-values indicate the statistical significance upon performing the annotated t tests.

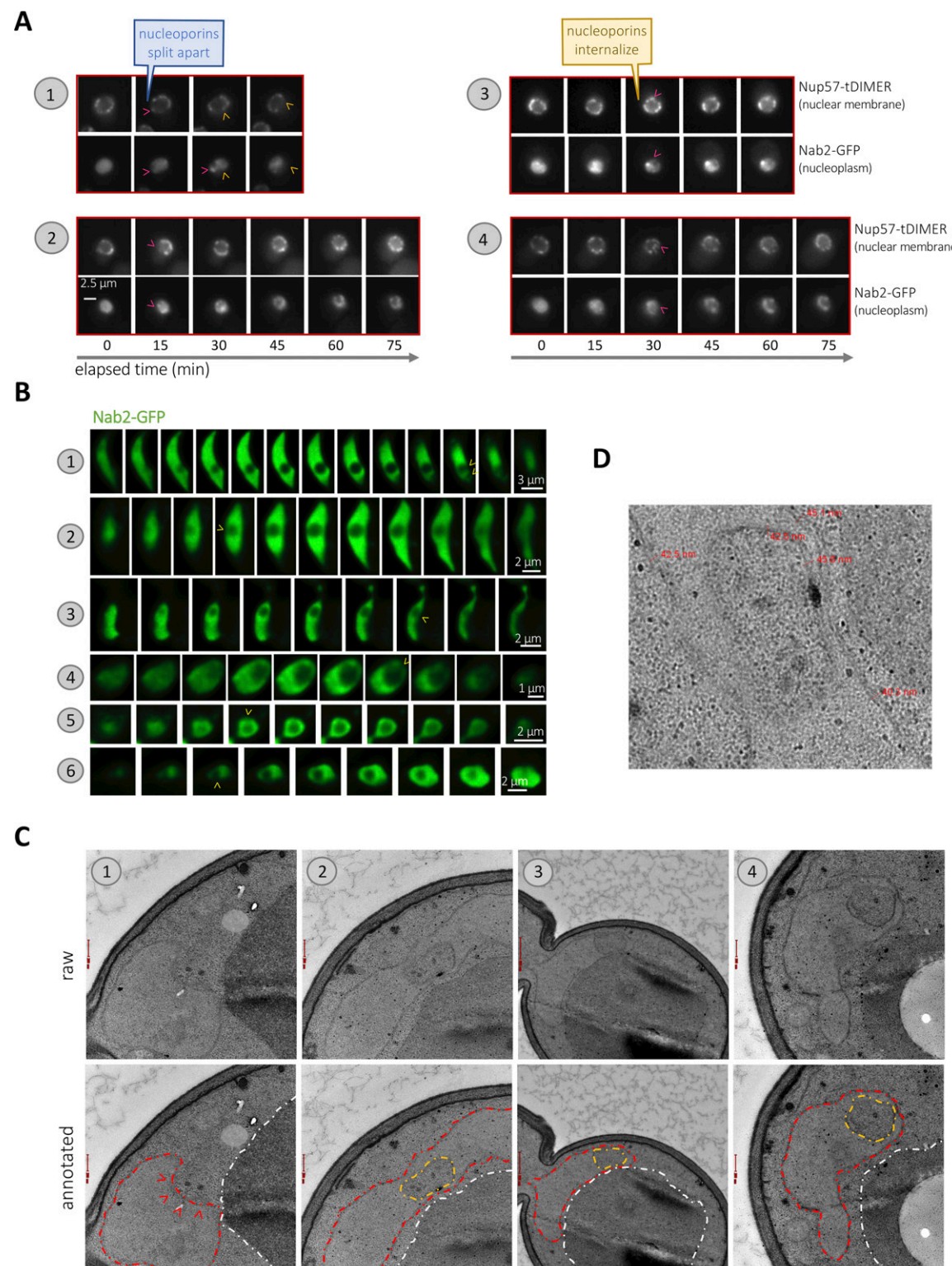

**Figure 3. Nuclear holes are globular cytoplasmic inclusions.**
**(A)** TIRF microscopy of exponentially growing *yeh2Δ* cells in which the nucleoplasmic protein Nab2 was labeled with GFP to monitor hole formation, and the nuclear envelope was monitored using Nup57-tDimer, were exposed to 100 µg/ml zeocin for 3 h and images acquired every 15 min for the indicated lapse of time at different positions of a FluoroDish plate. Arrowheads of different colors point at different internalization events. **(A, B)** Through-focus series of Nab2-GFP signals were acquired every 200 nm from cultures as described in (A). Numbers indicate six different examples. Yellow arrowheads point at the position where a connection of the hole with the cytoplasm could be perceived. **(C)** Electron microscopy sections of *steΔ* cells exposed to 100 µg/ml zeocin for 3 h before fixation. Different examples are indicated by numbers. Each image is shown twice: raw, and annotated, in this case with the nucleus artificially delimited by a dashed red line, the vacuole by a white dashed line and

induced nuclear holes being directly related to vacuoles (Fig 3C). To further explore this aspect, we used the vacuole membrane–specific dye MM4-64, which emits in the red wavelength range. Although part of the nuclear holes adjacent to or within the Nop1-CFP signals were vacuoles (Fig 5C), many were partially refractory to MM4-64 staining (Fig 5C, arrowheads). We also stained the vacuolar lumen using the pH-independent dye BCECF (Plant et al, 1999). This robust dye allowed us to detect eventual examples where the nuclear hole was irrefutably filled with BCECF signals (Fig 5D, yellow arrowheads), yet the prevalent scenario was that in which BCECF signals arising from nuclear holes were poor or lacking. Last, we used a GFP-tagged version of the vacuolar transmembrane protein Vph1 in combination with the mCherry-Pus1 tagging, which allows us to monitor nuclear holes. We almost never managed to observe vacuoles located inside the hole (Fig 5E). Thus, we conclude that the black holes detected in the nucleus, corresponding to portions of invaginated cytoplasm, eventually, but rarely, contain a "sequestered," trapped vacuole.

## Nuclear holes match a boost in autophagy

We hypothesized that, even if vacuoles themselves were rarely trapped within the nuclear holes, the fact that the ingression takes place at the nuclear envelope subdomain regularly used to establish contacts with the vacuole, could alter vacuole biology. We carefully re-analyzed our time-lapse Vph1-GFP mCherry-Pus1 microscopy experiments (Fig 5E) and realized that hole ingression systematically preceded the manifestation of what we call a "Vph1 signature" (Fig S3B). Vph1 is a key proton pump that acidifies the vacuole lumen, therefore conferring it its degradative potential, and is delivered to the vacuole membrane after originating in the Golgi (Manolson et al, 1992; Piper et al, 1997). We define the Vph1 signature as events in which Vph1-positive vesicles dock on a pre-existing vacuole, giving rise to a local (microdomain) increase in Vph1 signal intensity and to fusion-looking events (Figs S3B and 5E, arrowheads in the Vph1 channel). We counted and classified the cells considering whether they displayed a hole or not, and/or a positive Vph1 signature or not, creating with this information a 2 × 2 contingency table monitoring 659 cells (Fig S3C). A total of 78% of cells with a nuclear hole were also featured by a Vph1 signature, in contrast to 16% of Vph1 signature-positive cells in the absence of a hole. To provide a statistically reliable assessment of the apparent association between the presence of a nuclear hole and a Vph1 signature, we applied a $G$-test (Fowler et al, 1998). We calculated a $G_{adjusted}$ statistic (=119.627) that greatly exceeds that of the tabulated value of 6.63 at $P$ = 0.01 in a $\chi^2$-distribution at 1 degree of freedom, as corresponding in this case (Fowler et al, 1998). We therefore conclude that there is a highly significant association between the presence of the nuclear hole and that of a Vph1 signature.

Perhaps the local suction generated by the nuclear internalization of cytoplasmic portions in the vicinity of the vacuole creates a current, or a flow, that is favorable to the delivery of Vph1-containing vesicles to the vacuole membrane and/or to the cargoes approaching for degradation. With the vacuole being the central organelle where general autophagy takes place, we therefore hypothesized that hole formation may consequently impact autophagy. To monitor this, we used a broadly accepted tool consisting of the N-terminally GFP-tagged version of the auto-phagosome membrane-nucleating factor Atg8 (Geng et al, 2008; Nair et al, 2011). GFP-Atg8 molecules become degraded with the cargo, but the partial resistance to degradation of the GFP moiety permits the assessment of autophagy completion. This way, degradative vacuoles appear as green when scored by fluorescence microscopy. In addition, free GFP molecules, which migrate faster in a protein gel than the intact GFP-Atg8 ones, can be used to establish the percentage of degradation by Western blot. We compared the autophagic flux in conditions displaying increasing levels of nuclear holes (WT < $yeh2\Delta$ < WT + zeo < $yeh2\Delta$ + zeo, Fig 2G). We first treated, or not, WT and $yeh2\Delta$ cells with zeocin (zeocin effect was monitored by its ability to elicit the phosphorylation of the DNA damage response effector Rad53 [Fig S4A]) and induced autophagy 210 min later by adding rapamycin. Of note, the percentage of cells bearing at least one nuclear hole, and which continuously increased during zeocin exposure, was unaffected by rapamycin addition (Fig S4B). We observed a striking positive correlation between nuclear holes presence and the autophagic flux: the conditions triggering the highest number of cells with nuclear holes were the ones showing increased autophagic completion, irrespective of whether monitoring was done by Western blot (Fig 6A, % free GFP moieties) or by counting green vacuoles (Fig 6B). It is important to note that, although zeocin alone could elicit an increase in autophagy efficiency unrelated to hole formation (Eapen et al, 2017), the single absence of Yeh2 was sufficient to elicit this autophagy boost.

First, by definition, hole formation must be accompanied by an increase in nuclear membranes. In that case, a trivial explanation underlying the observed raise in autophagy would be that the increase in nuclear membranes provides more substrates to be degraded by autophagy. To assess this possibility, we repeated the monitoring of GFP-Atg8 degradation by comparing the ability in untreated versus zeocin conditions of a strain in which the Atg39 receptor, which instructs the autophagy of the nuclear membranes (Mochida et al, 2015; Otto & Thumm, 2021), had been removed. We observed that the incapacity to execute nuclear membrane autophagy still granted an accelerated autophagy completion when plus zeocin (Fig S4C). Thus, the enhanced autophagy execution we detect does not stem from an excess of nuclear membrane-derived cargoes.

Second, the increase in general autophagy completion could be arguably explained by an increase in the number of cargoes from any other cellular origin. To assess this, we counted the number of cells displaying GFP-Atg8 foci, which represent the incipient phagophores and the growing and mature autophagosomes before their fusion to and destruction within the vacuole. This way, one

---

the nuclear holes by an orange one. Arrowheads in "example 1" mark the invaginating membrane. Red bars are 200 nm-long except for example "3," where it is 500 nm-long. **(C, D)** Zoom of the example "2" presented in (C), rotated 90° left, displaying details and measurements of the width of the nuclear and the hole-delimiting membranes at three and two locations, respectively. Values oscillate between 40 and 45 nm.

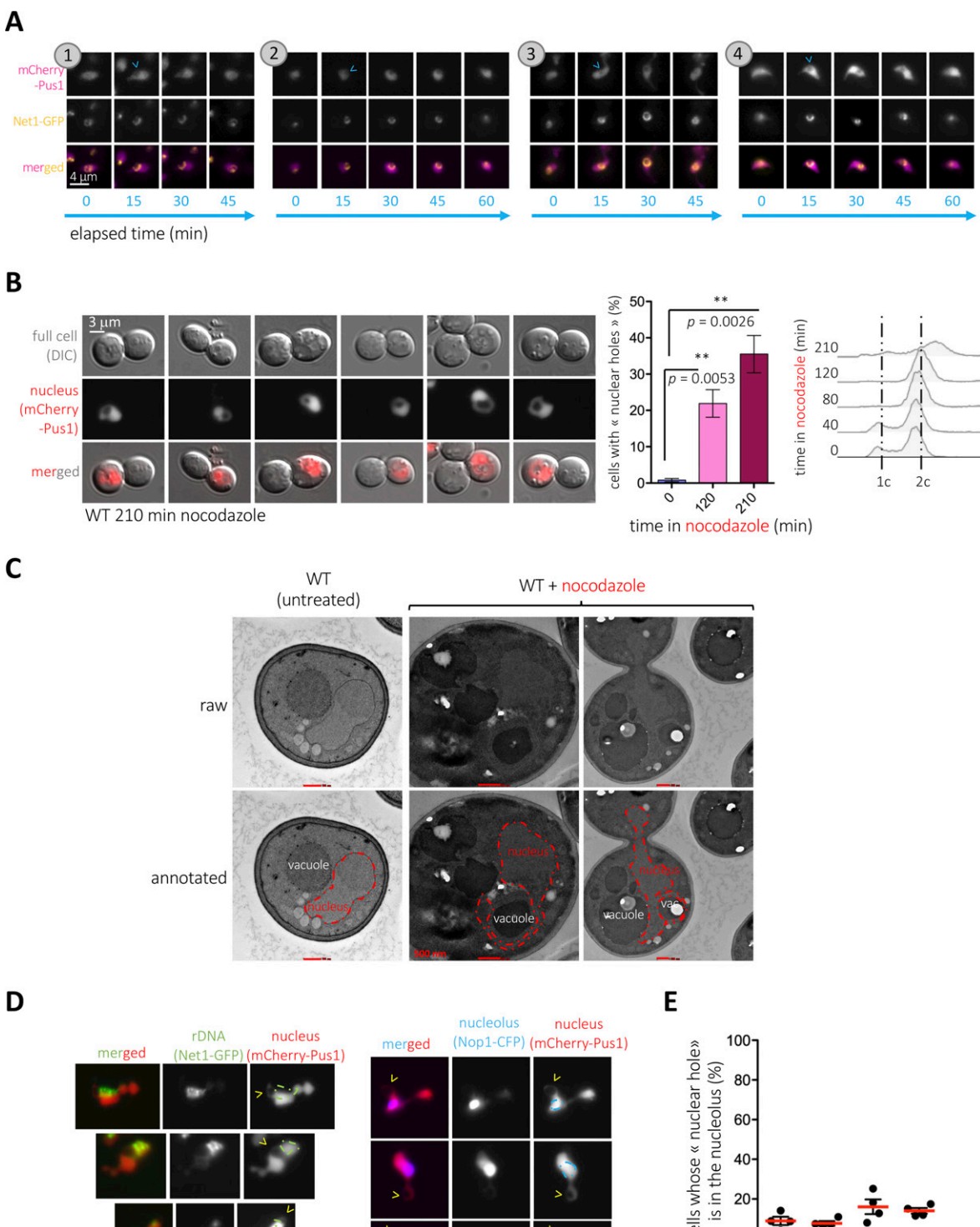

**Figure 4. The nuclear holes intrude from the nuclear envelope subdomain surrounding the nucleolus.**
**(A)** TIRF microscopy of exponentially growing *yeh2Δ* cells in which the rDNA-binding protein Net1 was labeled with GFP and the nucleoplasm with mCherry-Pus1 were exposed to 100 μg/ml zeocin for 3 h and images acquired every 15 min for the indicated lapse of time at different positions of a FluoroDish plate. Four different examples, indicated by numbers, are shown. Cyan arrowheads indicate the first detection of the nuclear hole. **(B)** WT cells bearing the mCherry-Pus1 construct were grown in a rich medium to the exponential phase and exposed to 15 μg/ml nocodazole for the indicated time. Cells were imaged and six examples are shown. The percentage of cells displaying nuclear holes was calculated and is plotted. The bars and the error bars are the mean and the SEM, respectively, of three independent experiments. At least 200 cells were counted per time-point, treatment, and experiment. The *P*-values indicate the statistical significance upon performing a *t* test. Cytometry profiles are shown,

would expect more foci in *yeh2Δ* than in WT cells if there were more cargoes to be degraded, whereas one would expect less foci in *yeh2Δ* than in WT cells if the increased autophagy completion was because of more efficient autophagosome clearance. Both when zeocin was added and when not, *yeh2Δ* cells recurrently displayed less GFP-Atg8 foci than WT cells (Fig S4D), arguing that the improvement in the ability of executing autophagy upon rapamycin addition stems from an increased vacuolar performance.

Last, the theoretical possibility existed that the cytoplasmic portions engulfed within the nucleus, whose aspect is reminiscent to a vacuole itself (Fig 6C, untreated, yellow arrows), could host degradative events that contribute to the detected raise in autophagy. To evaluate this, we transformed cells with a construct expressing both a pH-stable RFP, which testifies of cargo engulfment, and a pH-sensitive GFP, which reveals efficiency of degradation (see detailed explanation in the legend of Fig 6C and Rosado et al [2008]). The system was proficient because cargo engulfment in the vacuole was detected upon rapamycin addition (Fig 6C, rapamycin + CCCP, RFP channel, blue arrowheads), yet degradation was prevented by the protonophore CCCP (Fig 6C, rapamycin + CCCP, GFP channel, blue arrowheads) (Rosado et al, 2008). In agreement with proficient autophagy during zeocin treatment, cargoes were both engulfed and efficiently cleared when combining rapamycin and zeocin (Fig 6C, rapamycin + zeocin, diffuse RFP signal, poor GFP signal). Because the freely diffusing RFP and GFP signals concentrate within the constrained space of the nucleus, the presence of the holes was readily detectable as lighter regions. Yet, both signals were recurrently detected in the hole at pixel intensities similar to those measured in the cytoplasm (Fig 6D), indicating that the nuclear holes do not host any degradative events.

In conclusion, we propose that nuclear holes correspond to globular, engulfed portions of cytoplasm that occur in proximity to bona fide vacuoles. By a yet unknown mechanism, presumably related to the local depression created by the invagination process, they may favor Vph1 enrichment at or cargo delivery to the vacuolar membrane, thus endowing the cell with a stronger autophagic potential (Fig 7A, model).

## Discussion

In this work, we have identified a striking phenomenon through which the nuclear space is invaded by the ingression of globular bodies of cytoplasmic origin. This process, which mainly initiates at the nuclear envelope subdomain surrounding the nucleolus, necessitates the accumulation of PA and can be further fostered by high levels of free sterols. We also uncover its impact on the

efficiency of autophagy, which appears to concur with an improved delivery of the proton pump Vph1 and/or of autophagic cargoes to near-by vacuoles. Thus, we have uncovered a dramatic membrane-remodeling event with an immediate impact on metabolic adaptation.

We have serendipitously identified two situations in which, upon treatment of cells with two genotoxic agents, the phenomenon of cytoplasm internalization within the nucleus could be detected. It is hard to establish a common feature that can explain this because zeocin and MMS, the two triggering agents, do not create DNA damage in the same way. In this sense, zeocin provokes single and double DNA breaks, whereas MMS mostly alkylates DNA bases and therefore limits the use of DNA as a template. Furthermore, other genotoxins do not elicit the phenomenon under study (Figs 1B and S1B). Thus, we think we can safely say that the phenomenon is not related to DNA damage itself. The explanation could be temptingly related to the cell cycle phase because zeocin forces cells to arrest in $G_2$ (Fig 1C) and nocodazole, an agent forcing cells to accumulate in the $G_2$-to-M transition, also elicits a nuclear envelope extension phenotype (Fig 4) and (Matos-Perdomo et al, 2021 *Preprint*). That said, not all the treatments that lead cells cycle arrest in $G_2$ induce the formation of nuclear holes (Fig 1B and C), and we observed cells out of $G_2$ displaying nuclear holes (Fig 6C). We therefore favor the possibility that all the eliciting agents entail a stress provoking changes in the metabolism of lipids. In support, zeocin was reported to trigger membrane expansion (Witkin et al, 2012), methylglyoxal induces a lipid-driven nuclear deformation during vacuolar pushing (Nomura et al, 2020), and we recently reported that MMS elicits lipid alterations at the nuclear membrane (Ovejero et al, 2021; Kumanski et al, 2022).

In accordance, the mechanism we describe requests a raise in the level of PA or sterols. Given the negative curvature-imparting potential of these lipids (Fig 2B), this matches well our finding that the nuclear envelope becomes deformed toward the nucleoplasm. Yet, while we observe that, in response to zeocin, PA accumulates all over the INM and at discrete spots at the ONM (Figs 2E and F and S2C), and the nuclear envelope subdomain more prone to elicit the cytoplasmic ingression is mostly the one adjacent to the nucleolus (Fig 4A). What then does bookmark this region? The answer may reside in recent evidence revealing that key metabolic decisions downstream of PA, and essential for membrane remodeling, take place specifically at the nucleolus-surrounding membrane subdomain (Barbosa et al, 2019). Another feature of this membrane subdomain is that, as a rule, its zipper-wise contact with the vacuole increases to promote (piecemeal) autophagy of the nucleus (Kvam & Goldfarb, 2007; Mostofa et al, 2018; Mostofa et al, 2019). Linking both works, deregulations of PA at this location, as the excess of PA seen in the absence of the phosphatase Nem1, hampers autophagy

---

where 1c and 2c indicate the DNA content. **(C)** Electron microscopy sections of WT cells treated (or not) with nocodazole as described in (B). Each image is shown twice: raw, and annotated, with the nuclear-hole structure artificially delimited by a red dashed line. Each red bar corresponds to 500 nm. **(D)** The strain *yeh2Δ*, which displays nuclear holes at high frequency in response to 100 μg/ml zeocin (Fig 2G), was transformed either with a vector expressing Net1-GFP, to mark the position of the ribosomal DNA, or with a vector expressing Nop1-CFP, to mark the position of the nucleolus, grown to the exponential phase and treated with that drug. The comparison with nucleoplasmic mCherry-Pus1 signals allows monitoring the relative position of the nuclear holes, further highlighted by yellow arrowheads, with respect to these subnuclear domains. To facilitate visualization, the contour of the rDNA or the nucleolar signals has been over-imposed onto the nucleoplasmic ones. **(E)** The indicated strains, transformed with the vector expressing Nop1-CFP, were monitored for the relative position of the nuclear holes with respect to the nucleolus. The percentage of cells in the population in which the nuclear hole disrupted the nucleolus was counted. The red bar reflects the mean of four independent experiments, and the error bars account for the SEM. At least 200 cells were considered per condition and experiment.

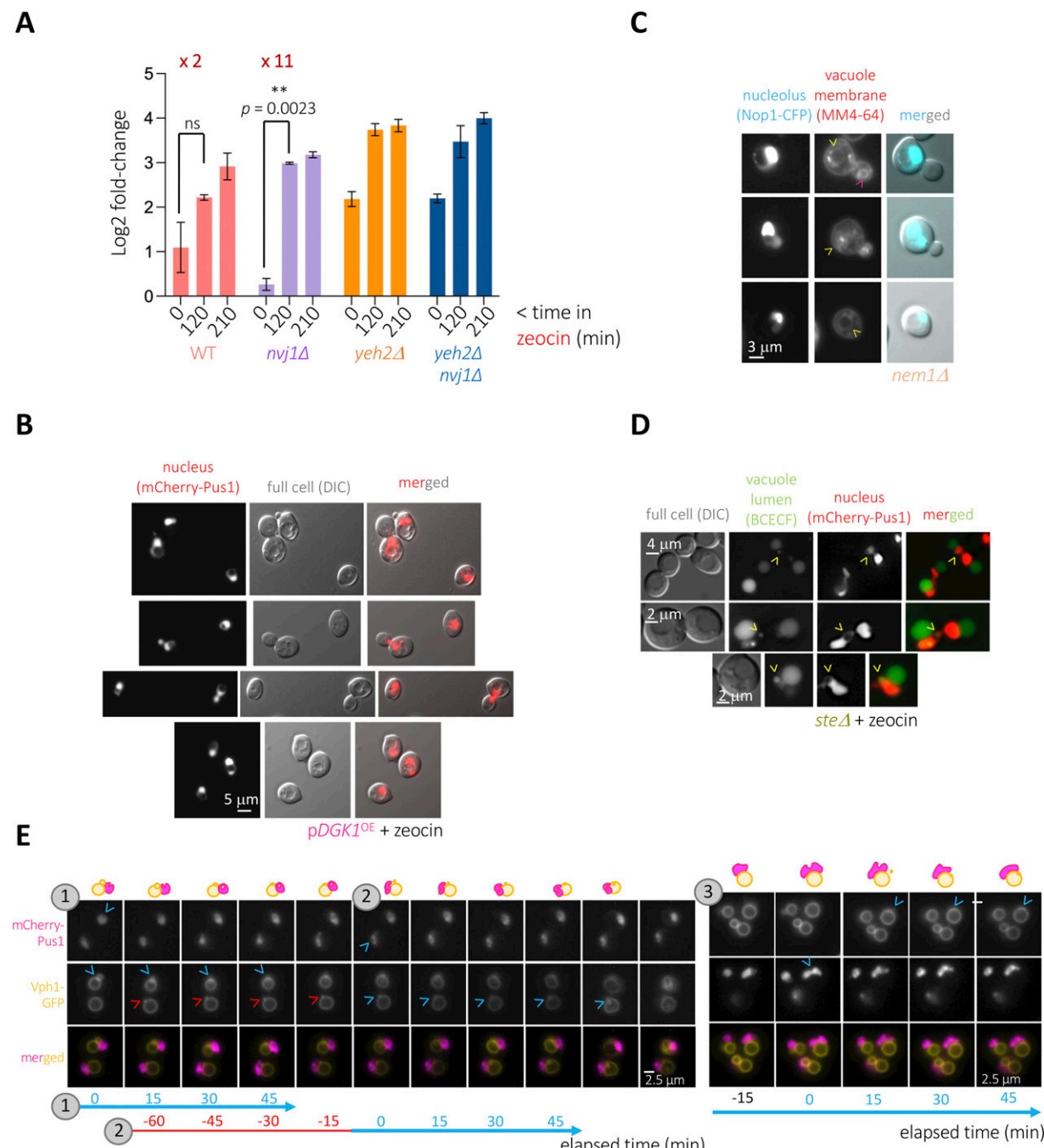

**Figure 5. Relationship between nuclear holes and vacuoles.**
**(A)** Cells of the indicated genotypes bearing the genomic mCherry-Pus1 construct and growing exponentially in the rich medium were photographed before 100 μg/ml zeocin addition and 120 and 210 min later. The percentage of cells in the population displaying at least one nuclear hole was counted. To facilitate comparison, the $\log_2$ of the fold-change (with respect to the lowest untreated WT value) has been plotted. Each bar reflects the mean of three independent experiments, and the error bars account for the SEM. At least 200 cells were considered per condition and experiment. The *P*-value indicates the statistical significance upon performing a paired *t* test. ns, nonsignificant. **(B)** Selected examples of cells in which the comparison of the mCherry-Pus1 signals and the position of the vacuoles, as seen from the differential interference contrast (DIC) images, permit to infer that the nuclear holes correspond to the vacuole. **(C)** The *nem1Δ* strain, transformed with the vector expressing Nop1-CFP, was simultaneously dyed with the vacuole membrane marker MM4-64. Nop1-CFP signals are overexposed to allow the visualization of the nuclear hole, mostly present in the non-Nop1–marked part of the nucleus. The MM4-64 signal coming from the hole-residing vacuole is poor (yellow arrowheads) and contrasts with that of the MM4-64 signal coming from cytoplasmic vacuoles (pink arrowhead). **(D)** The strain *steΔ*, which displays mCherry-Pus1–defined nuclear holes at high frequency in response to 100 μg/ml zeocin (Fig 2G), was grown in a rich medium to the exponential phase and treated with this drug for 3 h. Vacuole lumens were dyed using the dye BCECF and cells immediately imaged. Yellow arrowheads point at nuclear holes dyed with the BCECF marker. **(E)** TIRF microscopy of exponentially growing *yeh2Δ* cells in which the vacuolar membrane protein Vph1 was labeled with GFP and the nucleoplasmic Pus1 was marked with mCherry. Cells were exposed to 100 μg/ml zeocin for 3 h, and images acquired every 15 min for the indicated lapse of time at different positions of a FluoroDish plate. Three representative kinetics are shown. Time, indicated at the bottom

(Rahman et al, 2018a, 2018b). We observe that, in *nem1Δ* cells and upon Dgk1 overexpression, the nuclear envelope invagination is such that it engulfs the vacuole (Fig 5A and B), an extreme case in which the overall negative effect on autophagy could be explained because cargoes may experience difficulties to dock on the vacuole. Our data now raise an additional set-up where the nucleolus-surrounding membrane is shared between (1) establishing contacts with the vacuole and (2) supporting the ingression of cytoplasmic portions. We propose that this cytoplasmic engulfment could create a suction phenomenon (Fig 7A, "1") that locally accelerates the flow of surrounding material therefore fostering the delivery of the transmembrane proton pump Vph1 and/or of autophagic cargoes to the vacuole membrane (Fig 7A, "2"), which would underlie the observed increase in autophagy potential.

With respect to autophagy, it is also worth discussing the possible conservation of this phenomenon. The vacuole fulfills in *S. cerevisiae* the function the lysosomes accomplish in most animal and vegetal cells. In these cells, the proximity of lysosomes with the nucleus dictates the cell's ability to complete autophagy. As such, although autophagosomes form randomly at different locations within the cytoplasm, active lysosomes reside at the perinuclear region, where microtubules transport autophagosomes for fusion and subsequent autophagy (Kimura et al, 2008). Thus, lysosome position affects its acidity and therefore the cell autophagic potential (Korolchuk et al, 2011; Gowrishankar & Ferguson, 2016; Johnson et al, 2016). Of note, low cholesterol at membranes prevents this transport therefore decreasing autophagy execution (Wijdeven et al, 2016). Together, a common picture emerges in which, the higher the proximity with the nucleus (in the case we describe, proximity being favored by cytoplasmic suction within the nucleus), the higher the overall efficiency in autophagy, with both phenomena enhanced by high levels of free sterols in membranes. The analogy is further supported by a very recent work in which cytoplasmic portions harboring lysosomes accumulate within nuclear holes in late stages of the cell cycle in human cells (Almacellas et al, 2021).

Last, the intrusion in the nucleoplasmic space of a voluminous body is akin to alter nuclear processes. In a passive manner, just because of the space it occupies, it is likely to disrupt chromosome territories and displace, literally pushing, chromatin. Pushed chromatin behaves as condensed or compacted and as such will emit related signals (Burgess et al, 2014), impacting DNA transactions as transcription or repair. In agreement, the constriction of nuclei, and the chromatin landscape alterations this provokes, for example, in the shape of lobes, is of physiological relevance for the correct function of immune cells as the neutrophils (Georgopoulos, 2017; Zhu et al, 2017; Denholtz et al, 2020). Furthermore, the inclusion of a globular, yet still connected, portion of cytoplasm within the nucleus is a specific feature of a precise subtype of T-cell lymphoma, namely of anaplastic large cell lymphoma (ALCL) and a sporadic feature in other hematological diseases (Langenhuijsen, 1984; Kanoh et al, 1986; Lee et al, 1997; Brito Nascimento et al, 2020).

Although otherwise morphologically heterogenous, the universal presence of this type of nuclei in ALCL patients' samples has been established as their golden identification standard (Benharroch et al, 2012). This shape is strikingly reminiscent to the phenomenon we describe in this work: depending on the plane of cut, the nucleus will appear as a horseshoe (Fig 7B, "a"), as a canonical, round nucleus (Fig 7B, "b") or as a donut-like ring of different diameters (Fig 7B, "c," "d"). To go on with the analogy, and even if there is no evidence yet that this nuclear shape plays a specific role in the malignancy, it is worth of mention that ALCL cells are auxotrophic for cholesterol (Garcia-Bermudez et al, 2019), rely on the PA-producing enzyme DGK1 *α* (Bacchiocchi et al, 2005), and display an increased autophagy flux that is cytoprotective (Mitou et al, 2015). In this respect, the fact that the autophagy rate we see in the WT strain can be further improved by increasing sterols in membranes (Fig 6A and B, *yeh2Δ* versus WT) suggests that autophagy is basally "dampened" in the WT strain and that there exists a window for improvement of the autophagic capacity. This may relate to why inhibiting autophagy in ALCL proves of therapeutic potential (Mitou et al, 2015). Thus, our work may help shed light onto ALCL pathophysiology, particularly, and other hematological malignancies, more generally.

# Materials and Methods

### Cell culture and treatments

*S. cerevisiae* cells were grown at 25°C in YEP (rich) or yeast nitrogen base (YNB) (minimal) liquid medium supplemented with 2% glucose (dextrose), unless otherwise indicated. Transformed cells were selected for plasmid maintenance in YNB–leucine or YNB–uracil medium overnight. The morning after, the exponentially growing cultures were diluted and grown for at least 4 h in rich medium to create the optimal conditions to induce the formation of the "nuclear holes," unless otherwise indicated. To induce the overexpression of *DGK1* and *PAH1-7A*, cells were grown overnight in YNB–leucine with 2% glycerol. Then, 2% galactose was added to exponentially growing cultures to induce their expression. The strains and the plasmids used in this study are referred to in Tables 1 and 2, respectively.

### Reagents

4-NQO (N8141; Sigma-Aldrich), rapamycin (HY-10219; Cliniscience), MM4-64 (SC-477259; Santa Cruz Biotechnology), BCECF (216254; Sigma-Aldrich), methyl metanosulfonate (MMS, 129925; Sigma-Aldrich), zeocin (R25001; Thermo Fisher Scientific), nocodazole (M1404; Sigma-Aldrich), hydroxyurea (HU, H8627; Sigma-Aldrich), camptothecin (CPT, C9911; Sigma-Aldrich), CCCP (C2759; Sigma-Aldrich), and polylysine (P8920; Sigma-Aldrich).

in blue, refers to one specific event (nuclear hole/Vph1 signature) illustrated above and indicated by blue arrowheads. If indicated in red, both the time and the arrowheads refer to a static event that will resume/take place later (again indicated in blue). A drawing schematizing the visualized events is included at the very top of each column to help monitor them.

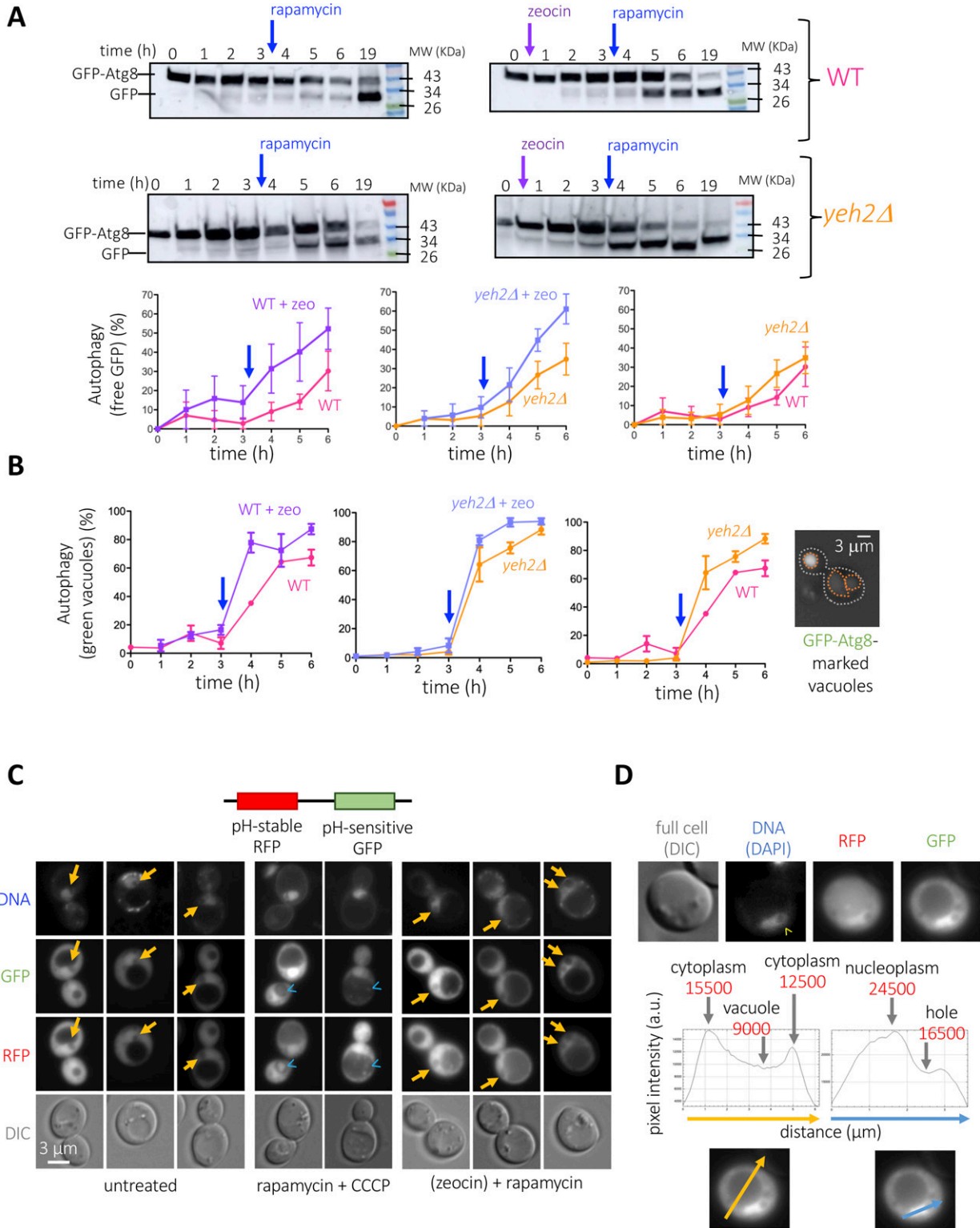

**Figure 6. Characterization of the relationship between nuclear holes and autophagy.**
**(A)** Cells of the indicated genotype, transformed with a vector expressing GFP-Atg8, were grown to the exponential phase in a rich medium and treated as indicated. 100 µg/ml zeocin was added or not, and 3 h later, 200 ng/ml rapamycin was added in all the cases. Samples were retrieved at the indicated time-points. The implementation of autophagy was monitored through Western blotting against GFP moieties. Time 19 h is included to illustrate that cells achieve a comparable level of autophagy. The quantifications plot the percentage of free GFP with respect to all the GFP signals in a given lane. The blue arrow is a reminder of the moment when rapamycin was added. The plotted points and the error bars are the mean and the SEM, respectively, of three independent kinetics. **(B)** The same experiment described in (A) was done, and cells were monitored by fluorescence microscopy. In this case, the level of autophagy was calculated as the percentage of cells in the population

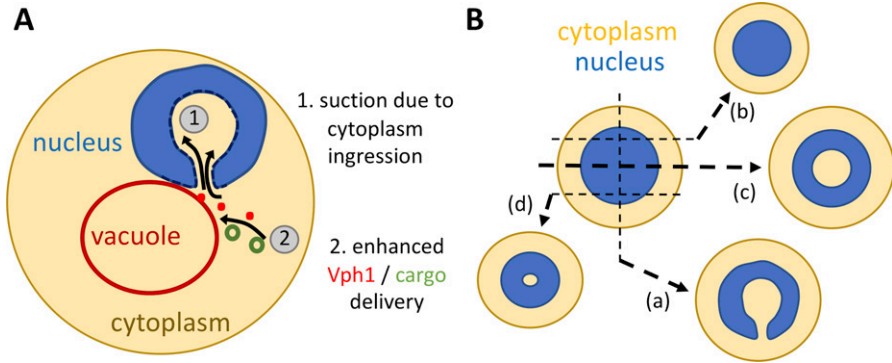

**Figure 7. Model.**
**(A)** Section of the cell in which the ingression of a globular cytoplasmic portion within the nucleus can be observed, a process that presumably triggers a phenomenon of local suction ("1"). This may attract thus concentrate Vph1-delivering entities (red dots) and/or cargoes targeted for degradation (green rings), which through more frequent docking and fusion with the vacuole membrane ("2"), will promote the increased autophagy potential. **(B)** Two concentric spheres represent the full cell (yellow cytoplasm) and the embedded nucleus (blue). Dashed, black lines represent cleaving planes. The putative outcomes of such slices are shown aside. In the event the cytoplasm invaginates in a globular fashion from the bottom of the drawn nucleus, the expected patterns can be kidney/horseshoe-like (a), canonically round (b), or donut-shaped profiles of variable diameters (c, d). The scheme is inspired from that presented in Pandiar and Smitha (2019).

## Cytometry

A total of 430 $\mu$l of culture samples at $10^7$ cells/ml were fixed with 1 ml of 100% ethanol. Cells were centrifuged for 1 min at 16,000$g$ and resuspended in 500 $\mu$l 50 mM Na-Citrate buffer containing 5 $\mu$l of RNase A (10 mg/ml, RB0474; Euromedex) for 2 h at 50°C. Then 6 $\mu$l of proteinase K (EU0090-C; Euromedex) was added for 1 h at 50°C. Aggregates of cells were dissociated by sonication (one 3 s-pulse at 50% potency in a Vibra-Cell 72405 Sonicator). Then 20 $\mu$l of this cell suspension was incubated with 200 $\mu$l of 50 mM Na-Citrate buffer containing 4 $\mu$g/ml propidium iodide (Thermo Fisher Scientific). Data were acquired and analyzed on a Novocyte Express (Novocyte).

## Protein extraction and Western blot

Approximately 5 × $10^8$ cells were collected at each relevant time point and washed with 20% trichloroacetic acid to prevent proteolysis, then resuspended in 200 $\mu$l of 20% trichloroacetic acid at 4°C. The same volume of glass beads was added, and cells were disrupted by vortexing for 10 min. The resulting extract was spun for 10 min at 1,000$g$ also at room temperature, and the resulting pellet resuspended in 200 $\mu$l of 2× Laemmli buffer. Whenever the resulting extract was yellow-colored, the minimum necessary volume of 1 M Tris base (noncorrected pH) was added till blue color was restored. Then, water was added till reaching a final volume of 300 $\mu$l. These extracts were boiled for 10 min and clarified by centrifugation as before. To separate Rad53 isoforms, 10–15 $\mu$l of this supernatant was loaded onto a commercial 3–8% acrylamide gradient gel (Bio-Rad) and migrated 70 min at 150 V in 1x Tris-acetate buffer. The same volume of supernatant was used to separate GFP from GFP-Atg8 isoforms onto a commercial 4–20% acrylamide gradient gel (Bio-Rad) and migrated 45 min at 100 V in 1× MES buffer. Proteins were transferred to a nitrocellulose membrane. Detection by immunoblotting was accomplished with the anti-Rad53 antibody (1/3,000), a kind gift from Dr. C. Santocanale, Galway, Ireland; or anti-GFP antibody (TP-401, 1/2,000; Clinisciences), respectively, and in both cases an anti-rabbit HRP secondary antibody (A9044-2ML, 1/3,000; Sigma-Aldrich).

## Standard fluorescence microscopy

First, 1 ml of the culture of interest was centrifuged; then, the supernatant was thrown away, and the pellet was resuspended in the remaining 50 $\mu$l. Next, 3 $\mu$l of this cell suspension was directly mounted on a coverslip for immediate imaging of the pertinent fluorophore-tagged protein signals. To dye vacuole membranes, 2 $\mu$l of a 4 mM MM4-64 stock was added to 1 ml of culture under incubation 30 min before visualization. To dye vacuole lumens, BCECF was added to and mixed with the the 50 $\mu$l of centrifuged pellet with the residual medium at a 50 $\mu$M final concentration immediately before mounting. Imaging was achieved using a Zeiss Axio Imager Z2 microscope, and visualization, co-localization, and inspection performed with ImageJ. Through-focus series were acquired every 0.26 for 8 $\mu$m. Deconvolution was done with Huygens professional v21.04 using the CMLE algorithm. 3D views reconstructions and slices were obtained using Imaris v9.8.

## Time-lapse microscopy

Yeast cells of the indicated phenotype were grown on liquid cultures in LoFlo medium (LFG0501; Formedium) supplemented with

---

displaying green vacuoles, indicative of autophagy completion. An example of such a cell is displayed on the right, with the cell contour drawn in white and the vacuole one in orange. The blue arrow is a reminder of the moment when rapamycin was added. The plotted points and the error bars are the mean and the SEM, respectively, of three independent kinetics. **(C)** Exponentially growing cells expressing the pRosella vector (Rosado et al, 2008) were either left untreated, or treated with 100 $\mu$g/ml zeocin for 3 h, then 200 ng/ml rapamycin added for three more hours. As a control, cells were treated with 200 ng/ml rapamycin alone for 4 h (autophagy induction) or with rapamycin plus 10 $\mu$M of the protonophore CCCP during the last hour as a control to block degradation within the vacuole of committed cargoes. The pRosella vector leads to the expression of a pH-stable RFP protein, which allows monitoring cargo engulfment within the vacuole, and a pH-sensitive GFP, which cannot fluoresce from degradation-competent vacuoles, whose pH is acidic, yet will emit from nonacidic environments. Yellow arrows indicate holes. Blue arrowheads point at engulfed yet undegraded cargoes. **(D)** One cell from a sample treated with zeocin then rapamycin and displaying a nuclear hole (yellow arrowhead) is shown in the four indicated channels. The image corresponding to the GFP channel (efficient autophagy detected as a dark vacuole) has been exploited to measure pixel intensity along a line that cuts the cytoplasm/the vacuole (yellow line) and the nucleus/nuclear hole (blue line). The aim was to unambiguously determine whether the signal intensities present in the nuclear hole resembled that of the same cell's cytoplasm, nucleus, and/or vacuole.

**Table 1.** Strains used in this study.

| Simplified genotype | Full genotype | Source |
|---|---|---|
| WT (background BY) | MAT a, his3Δ1, leu2Δ0, met15Δ0, ura3Δ0 | EUROSCARF |
| yeh2Δ (background BY) | MAT a, his3Δ1, leu2Δ0, met15Δ0, ura3Δ0, yeh2ΔG418$^R$ | Zvulum Elazar |
| Net1-GFP mCherry-Pus1 (background BY) | MAT a, his3Δ1, leu2Δ0, met15Δ0, ura3Δ0, yeh2ΔG418$^R$, NET1-GFP-LEU2, mCherry-PUS1::URA3 | MM-198, this study |
| Rad52-YFP Rfa1-CFP mCherry-Pus1 (background W303) | MAT a, ade2, his3, can1, leu2, trp1, ura3, RAD52-YFP RFA1-CFP mCherry-PUS1::URA3 | PP3558, Philippe Pasero |
| yEGFP-Tel1 mCherry-Pus1 (background W303) | MAT a, ade2, his3, can1, leu2, trp1, ura3, GAL+, psi+, RAD5+, yEGFP-TEL1, mCherry-PUS1::URA3 | MM-40 (Coiffard et al, 2021 Preprint) |
| atg39Δ (background W303) | MAT a, ade2, his3, can1, leu2, trp1, ura3, GAL+, psi+, RAD5+, atg39ΔG418$^R$ | MM-37, this study |
| Nup49-GFP (background W303) | MAT a, ade2, his3, leu2, trp1, ura3, ARS604::LacOp-TRP1, Nup49-GFP | PP3188, Armelle Lengronne |
| are1Δ are2Δ (steΔ) (background W303) | MAT alpha, ade2, his3, can1, leu2, trp1, ura3, are1ΔHIS3 are2ΔLEU2 | Zvulum Elazar |
| WT (background RS453) | MAT ?, ade2-1, his3-11,15, leu2-3,112, trp1-1, ura3-52 | SS2236, Symeon Siniossoglou |
| dgk1Δ (background RS453) | MAT ?, ade2-1, his3-11,15, leu2-3,112, trp1-1, ura3-52, dgk1Δ | SS1144, Symeon Siniossoglou |
| pah1Δ (background RS453) | MAT ?, ade2-1, his3-11,15, leu2-3,112, trp1-1, ura3-52, pah1Δ + pPAH1-URA3 | SS1746, Symeon Siniossoglou (cured from pPAH1-URA3 before experiments) |
| nem1Δ (background RS453) | MAT ?, ade2-1, his3-11,15, leu2-3,112, trp1-1, ura3-52, nem1Δ | SS1960, Symeon Siniossoglou |
| Vph1-GFP | MAT a, his3Δ1 leu2Δ0 met15Δ0 ura3Δ0, VPH1-GFP::HIS3 | Alenka Čopič (Huh et al, 2003) |
| Nab2-GFP Nup57-tDIMER (background W303) | MAT a, ade2, his3, leu2, trp1, ura3, NUP57-tDIMER::LEU2 NAB2-GFP::URA3 yeh2ΔG418$^R$ | MM-366, this study |
| Vph1-GFP mCherry-Pus1 | MAT a, ade2, his3, leu2, trp1, ura3, VPH1-GFP::HIS3 mCherry-PUS1::URA3 yeh2ΔG418$^R$ | MM-381, this study |
| nvj1Δ | MAT a, ade2, his3, can1, leu2, trp1, ura3, GAL+, psi+, RAD5+, mCherry-PUS1::URA3 nvj1ΔHPHMX6 | MM-431, this study |
| yeh2Δ nvj1Δ | MAT a, ade2, his3, can1, leu2, trp1, ura3, GAL+, psi+, RAD5+, yEGFP-TEL1 mCherry-PUS1::URA3 yeh2ΔG418$^R$nvj1ΔHPHMX6 | MM-417, this study |

2% glucose till the exponential phase, and then zeocin 100 µg/ml was added 1 h before microscopy to elicit the phenotype of interest. Cultures were briefly centrifuged at 3,000g, and then resuspended in 100 µl LoFlo medium. A total of 10 µl were spotted on the bottom of a previously prepared FluoroDish (LoFlo rich medium supplemented with 2% glucose, 100 µg/ml zeocin, and 1% agarose was casted on FluoroDishes [FD35-100; World Precision Instruments] that had been previously treated with a 0.1% [wt/vol] polylysine solution for 10 min at room temperature, then air-dried). Samples were observed in a Metamorph-controlled Nikon TIRF PALM STORM microscope in a culture chamber at 30°C. Lasers were used at 5% of their power, and pictures were shot every 15 min.

### Electron microscopy

Cells were immersed in a solution of 2.5% glutaraldehyde in 1× PHEM buffer (pH 7.4) overnight at 4°C. They were then rinsed in PHEM buffer and post-fixed in 0.5% osmic acid + 0.8% potassium hexacyanoferrate trihydrate for 2 h in the dark at room temperature. After two rinses in PHEM buffer, the cells were dehydrated in a graded series of ethanol solutions (30–100%). Cells were embedded in EmBed 812 using an Automated Microwave Tissue Processor for Electronic Microscopy, Leica EM AMW. Thin sections (70 nm; Leica-Reichert Ultracut E) were collected at different levels of each block. These sections were counterstained with 1.5% uranyl acetate in 70% ethanol and lead citrate and observed using a Tecnai F20 transmission electron microscope at 120 KV located in the Institut des Neurosciences de Montpellier, Electronic Microscopy facilities, INSERM U1298, Université Montpellier, Montpellier France.

### Quantification of Western blots

ImageJ was used to determine the pixel intensity values associated with the two bands (GFP-Atg8 and GFP) present in each lane. The percentage of autophagy was calculated by dividing the signal associated to free GFP divided the total signal measured in the lane, multiplied by 100.

**Table 2.  Plasmids used in this study.**

| Simplified name | Detailed information | Source |
|---|---|---|
| pEmpty (-ura) | pRS316 | Benjamin Pardo (Sikorski & Hieter, 1989) |
| pEmpty (-leu) | YEplac181 | Symeon Siniossoglou (O'Hara et al, 2006) |
| pDGK1 | YEplac181-GAL1/10p-DGK1 | Symeon Siniossoglou (Karanasios et al, 2013) |
| pPAH1-7A | YEplac181-GAL1/10p-PAH1-7A | Symeon Siniossoglou (O'Hara et al, 2006) |
| pNOP1-CFP | pNOP1-CFP::LEU2 | Danesh Moazed (Mekhail et al, 2008) |
| pNET1-GFP | pDM266 (pNET1-GFP-LEU2) noncentromeric, digested with BglII allows integration within the endogenous NET1 locus | Félix Machín (Matos-Perdomo & Machín, 2018) |
| pNLS-Q2-mCherry | pRS316-CYC1p-Nup60$^{1-24}$(NLS)-Opi1$^{Q2}$-mCherry-NUP1t | Alwin Köhler (Romanauska & Köhler, 2018) |
| pGFP-Atg8 | pGFP-ATG8-URA3 | Wei-Pang Huang (Wang et al, 2015) |
| pAS1NB c Rosella I | Biosensor comprised of a fast-maturing pH-stable RFP fused to a pH-sensitive GFP | Addgene #71245 (Rosado et al, 2008) |
| pNAB2-GFP | ZJOM18 (pNAB2-GFP-URA3) integrative vector, digested with KasI allows integration at endogenous NAB2 locus | Addgene #133631 (Zhu et al, 2019) |
| pQ2-mCherry | pRS316-CYC1p-Opi1$^{Q2}$-mCherry-NUP1t | pMM-13 (Kumanski et al, 2022) |
| pNUP57-tDIMER | pRS305-NUP57-tDIMER | Olivier Gadal (Wang et al, 2016) |

ALCL, anaplastic large cell lymphoma; BCECF, 2′,7′-bis-(2-carboxyethyl)-5-(and-6)-carboxyfluorescein acetoxymethyl ester; CCCP, carbonylcyanure *m*-chlorophénylhydrazone; CFP, cyan fluorescent protein; CPT, camptothecin; DAG, diacylglycerol; DIC, differential interference contrast; GFP, green fluorescent protein; HU, hydroxyurea; INM, inner nuclear membrane; MMS, methylmethane sulfonate; NLS, nuclear localization signal; NVJ, nucleus–vacuole junction; ONM, outer nuclear membrane; PA, phosphatidic acid; rDNA, ribosomal DNA; SEM, standard error of the mean; TIRF, total internal reflection fluorescence; WT, wild type; 4-NQO, 4-nitroquinoline 1-oxide.

## Quantification of images

The determination of the percentage of cells in the population displaying nuclear holes was done by visual counting by the experimenter. Three independent experimenters participated in this counting as to warrant reproducibility and reliability.

## Graphical representations and statistical analyses

Graphical representations and statistical analyses were made with GraphPad Prism to both plot graphs and statistically analyze the data. For data representation, the SEM was used. The SEM estimates how far the calculated mean is from the real mean of the sample population, whereas the SD informs about the dispersion (variability) of the individual values constituting the population from which the mean was drawn. As all the measurements we were considering for each individual experiment concerned a mean (the percentage of cells in the population presenting nuclear "holes") and the goal of our error bars was to describe the uncertainty of the true population mean being represented by the sample mean, we did the choice of plotting the SEM. To assess whether nuclear hole and Vph1 signature presences were statistically significantly associated, data were organized as a 2 × 2 contingency table and a G-test applied, as advised (Fowler et al, 1998).

## Data Availability

All relevant data supporting this work are included in this manuscript. Any additional request of information will be satisfied upon direct contact with the corresponding author.

## Supplementary Information

## Acknowledgments

We are very thankful to Sebastian Schuck for the gift of pSec63-GFP; to Symeon Siniossoglou for the plasmid allowing mCherry-Pus1 tagging; the vectors to overexpress Dgk1 and Pah1-7A; and the *dgk1Δ*, *nem1Δ*, and *pah1Δ* strains and to Zvulum Elazar for *steΔ* and *yeh2Δ* strains. We also thank Alwin Köhler for pNLS-Q2-mCherry, Félix Machín for pNet1-GFP, Danesh Moazed for pNop1-CFP, and Alba Torán-Vilarrubias for creating the *atg39Δ* strain; Corrado Santocanale for the kind gift of anti-Rad53 antibody; Pr. Wei-Pang Huang for the present of pGFP-Atg8; and Philippe Pasero and Benjamin Pardo for the Rad52-YFP Rfa1-CFP strain and pRS316, respectively. We are very grateful to Florence Gaven for suggesting us the use of LoFlo medium and sharing tips for time-lapse experiments and to Volker Bäcker for invaluable training on how to use Huygens and Imaris. We are indebted to Simonetta Piatti for tips on time-lapse microscopy and critical reading of the manuscript, to Félix Machín for sharing nonpublished information and critical reading of the manuscript, and to Lucile Espert for critical reading of the manuscript and helpful insights to improve this work. We acknowledge the imaging facility MRI, a member of the national infrastructure France-BioImaging, supported by the French National Research Agency (ANR-10-INBS-04, Investissements d'avenir). We also thank the joint IGMM-CRBM "yeast media and technologies service" for providing us with ready-to-use media. We finally thank the ATIP-Avenir program, La Ligue contre le Cancer et l'Institut National du Cancer (PLBIO19-098 INCA_13832), France, for funding our research.

### Author Contributions

M Garcia: conceptualization, formal analysis, validation, investigation, methodology, and writing—review and editing.

S Kumanski: conceptualization, data curation, formal analysis, validation, investigation, methodology, and writing—review and editing.

A Elias-Villalobos: conceptualization, data curation, formal analysis, methodology, and writing—review and editing.

C Cazevieille: formal analysis, investigation, methodology, and writing—review and editing.

C Soulet: data curation, formal analysis, and writing—review and editing.

M Moriel-Carretero: conceptualization, data curation, supervision, funding acquisition, validation, investigation, visualization, methodology, project administration, and writing—original draft, review, and editing.

## Conflict of Interest Statement

The authors declare that they have no conflict of interest.

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
