## [Reviewer comments · Life Science Alliance]

Life Science Alliance

Nuclear ingressión of cytoplasmic bodies accompanies a boost in autophagy

Manon Garcia, Sylvain Kumanski, Alberto Elías-Villalobos, Chantal Cazevieille, Caroline Soulet, and María Moriel-Carretero
DOI: <https://doi.org/10.26508/lsa.202101160>

Corresponding author(s): María Moriel-Carretero, Centre de Recherche en Biologie cellulaire de Montpellier (CRBM), Université de Montpellier, Centre National de la Recherche Scientifique, Montpellier, France

Review Timeline:

Submission Date:	2021-07-15
Editorial Decision:	2021-09-08
Revision Received:	2022-01-17
Editorial Decision:	2022-02-18
Revision Received:	2022-04-22
Editorial Decision:	2022-04-26
Revision Received:	2022-05-02
Accepted:	2022-05-02

Scientific Editor: Novella Guidi

Transaction Report:

September 8, 2021

Re: Life Science Alliance manuscript #LSA-2021-01160-T

Dr. María Moriel-Carretero
Centre de Recherche en Biologie cellulaire de Montpellier (CRBM)
1919 Route de Mende
Montpellier 34293
France

Dear Dr. Moriel-Carretero,

Thank you for submitting your manuscript entitled "Negative curvature-promoting lipids instruct nuclear ingression of low autophagic potential vacuoles". The manuscript has been evaluated by expert reviewers, whose reports are appended below. Unfortunately, after an assessment of the reviewer feedback, our editorial decision is against publication in Life Science Alliance.

Although your manuscript is intriguing, I feel that the points raised by the reviewers are more substantial than can be addressed in a typical revision period. If you wish to expedite publication of the current data, it may be best to pursue publication at another journal.

Given the interest in the topic, I would be open to resubmission to Life Science Alliance of a significantly revised and extended manuscript that fully addresses the reviewers' concerns and is subject to further peer-review. If you would like to resubmit this work to Life Science Alliance, please contact the journal office to discuss an appeal of this decision or you may submit an appeal directly through our manuscript submission system. Please note that priority and novelty would be reassessed at resubmission.

Regardless of how you choose to proceed, we hope that the comments below will prove constructive as your work progresses. We would be happy to discuss the reviewer comments further once you've had a chance to consider the points raised in this letter.

Thank you for thinking of Life Science Alliance as an appropriate place to publish your work.

Sincerely,

Reviewer #1 (Comments to the Authors (Required)):

The study by Garcia et al is based on the observation that in the presence of reagents that lead to a G2/M arrest in budding yeast, there is an area in the nucleus that does not contain a nucleoplasmic marker (henceforth nuclear "hole", as the authors refer to it). The authors further show that these holes coincide with a dysfunctional vacuole. Using fluorescence imaging, the authors conclude that these holes are internalized vacuoles. If true, this is a remarkable observation, as an internalization of a vacuole into the nucleus has not been reported before, as far as this reviewer knows. Thus, the significance of this study rests on the quality of the data supporting this claim.

Previous work from several labs showed that during a G2/M cell cycle delay, the nucleus forms a loop-shaped protrusion (for example, Witkin et al 2012). Moreover, a recent study posted on BioRxiv from the Machin lab showed that this loop surrounds the vacuole. Thus, if the holes observed here are the same as the space associated with the nuclear envelope loops, the simplest explanation is that the holes are adjacent to the nucleus, or perhaps even engulfed by the nucleus, rather than internalized. The distinction between these two possibilities is straightforward: if the hole/vacuole is adjacent to the nucleus, the hole/vacuole lumen would be separated from the nucleoplasm by three membranes (the vacuolar membrane and the two nuclear membrane). If, however, the hole/vacuole is internalized, the nuclear membranes should be absent from the vacuole's periphery.

The present study has two major issues: first, the characterization of the spatial relationship between the hole/vacuole and nucleus is insufficient, and consequently the conclusion that the hole reflects vacuoles that are completely internalized by the nucleus, as the authors suggest, is premature. Second, the authors are presenting an endpoint condition. Live imaging would allow the determination of how the vacuole becomes surrounded/engulfed/internalized by the nucleus, and when during this

process the vacuole becomes dysfunctional. The comments listed below are aimed at addressed these and other issues. As it currently stands, the authors are grossly over-interpreting their data.

Major issues:

1. The main issues with this study are the lack of appropriate nuclear envelope and vacuole membrane markers and the absence of 3D reconstructions of both nuclei and vacuoles using, for example, confocal microscopy. This would enable the authors to distinguish between the two possibilities noted above (namely adjacent/engulfed vs internalized). Ideally this would be done by EM, but for an initial study fluorescence microscopy could suffice. It is quite possible that the vacuole is surrounded by the thin nuclear envelope protrusion (think shoelace around a ball) rather than surrounded by nuclear material from all sides (i.e. engulfed or internalized). This would also overcome the difficulties the authors have in observing the vacuole and will allow them to do live imaging so that they can follow the process by which the nucleus surrounds the vacuole as the cells become arrested in G2/M.
2. Figure 2A-D is (a) not novel (see for example Han et al, JBC 2008 and other studies for the effect of DGK1 over expression and Pah1 and its various mutations on nuclear morphology) and (b) over interpreted, because the authors do not know that PA of DAG are the causative agents of the nuclear hole- they can be metabolized into something else (for example, phospholipids) that is needed for membrane proliferation, for example.
3. The authors conclude that PA accumulates in the inner nuclear membrane, in support of providing negative membrane curvature driving this membrane to bend in a concave fashion. However, their reporter (NLS-PA-sensor-mCherry) was directed to the nucleoplasm, which precluded them from examining whether PA is equally accumulating on the outer nuclear membrane, invalidating their model. This experiment should either be eliminated or carried out with the appropriate sensors to detect PA (or lack thereof) on the outer nuclear membrane.
4. To quantify the spatial relationship between the holes/vacuoles and the nucleolus, the authors should repeat Figure 3 using a fluorescence vacuolar membrane marker (e.g. Vph1-GFP) in addition to the nucleolar marker (and preferably also a nuclear marker) and carry out 3D reconstructions based on confocal images. This is an important result because if the holes are not near the nucleolus, that will distinguish them from the mitotic nuclear membrane extensions described previously. Note, however, that yeast may have multiple nuclei and there are likely many vacuoles that are not associated with the nucleus anyway. When conducting these experiments, it will be beneficial to account for all cellular vacuoles.
5. What fraction of the total vacuolar volume is present in the nuclear hole? According to Figure 3E, where the authors used a vacuolar lumen marker, only a tiny fraction of the total vacuolar mass is nuclear-associated. If this is the case, it's not clear how the dysfunction of this small portion has such a large effect on autophagy (Figure 4). Is it possible that the effect of genotoxic agents on autophagy is a general one, not related to the specific nuclear-associated vacuole? Also, the authors state "a striking correlation between nuclear hole presence and autophagic flux. However, they did not examine nuclear holes in the presence of rapamycin, which is the condition under which autophagy was examined. Rapamycin is known to inhibit phospholipid synthesis and it is quite possible that under these conditions the number of holes is significantly diminished.
6. It is interesting that the vacuoles that are associated with the nucleus are less functional, but unless the authors can demonstrate unequivocally that the holes/vacuoles are inside the nucleus, the section on "vacuole sequestration" is grossly over-interpreted and should be removed or re-written (including Figure 5)

Minor issues:

1. The title of the manuscript is an over/mis-interpretation of the data does not reflect the novel findings in the study.
2. Please provide representative images to go with Figure 2A so that the reader can evaluate the morphology of the nuclei under the various conditions.
3. It is not clear what the authors meant by embedded sterols can act like adjuvants".
4. Figure 3D: please use fluorescently tagged nuclear envelope markers (NPC subunits, LINC complex, Heh1, etc) rather than an ER marker in conjunction with a fluorescently tagged vacuolar membrane marker (e.g. VPH1) and image the cells using confocal/high-resolution microscopy.

Reviewer #2 (Comments to the Authors (Required)):

Review comment:

This paper by Garcia et al. explores an unusual phenotype in budding yeast where there is a "hole" in the nucleus that is visualized by light microscopy using a nucleoplasmic reporter following treatment of cells with genotoxins. The authors make the provocative conclusion that this hole represents a vacuole that has been fully internalized into the nucleus. The authors demonstrate that the formation of the nuclear hole requires phosphatidic acid (PA) through several different genetic perturbations and by directly visualizing PA at the INM using a genetically encoded fluorescent PA sensor. They further provide evidence that such vacuole internalization might limit autophagy flux. Overall, the data are of good quality and investigate a unique and potentially interesting phenomenon that has not been documented in the past. Thus, there is the potential that new biology will stem from these foundational observations. Were the data to stand solely on their characterization and preliminary functional analysis of this phenotype, I would be supportive of publication in its current form, however, the authors choose a very provocative conclusion (that the vacuole is somehow internalized into the nucleus), which is not supported by the data and

would require substantial experiments to solidify.

Major Point:

1) Although the authors seemingly observe vacuoles in the nucleus, this conclusion is reliant on wide-field fluorescence microscopy, which has insufficient resolution to rule out that vacuoles might simply be surrounded by a nuclear envelope lobulation instead of being fully internalized. The suggestion is to substantially tone down the conclusions of the manuscript in this respect and to consider more likely explanations (see point 2, below). The title would also need to be changed. If the authors choose to be more definitive experimentally, they will need substantial high quality ultrastructural data that would really need to be serial section EM or FIB-SEM in order to capture the entire nuclear contour. In addition, or alternatively, they could provide direct evidence that the internalized vacuole is biochemically segregated from the cytoplasm.

Additional Points for authors/editors to consider to strengthen the manuscript:

2) Given the requirement for PA and the proximity of the nuclear holes to the nucleolus (which is established to be prone to membrane expansion upon perturbation to PA metabolism), the authors should consider the possibility that the hole they see results from an extended nuclear flare as defined by work from Orna Cohen-Fix. A more careful quantification of the proximity of the "hole" to the nucleolus would seem warranted. The authors could also consider performing time-lapse microscopy following zeocin treatment in the context of NE, nucleolar and vacuolar labels to understand the biogenesis of these structures more fully.

3) The authors' conclusion that the increase in autophagic flux following zeocin treatment is due to the internalization of low autophagic potential vacuoles rather than a general increase in autophagy caused by the zeocin itself needs to be tested more rigorously. The idea that vacuoles are not identical in their potential to degrade autophagic cargo is interesting and testable. As a tool, I suggest looking at Atg8 tagged with a tandem GFP-RFP tag utilizing a pH sensitive GFP (e.g. the Rosella tag, Rosado et al. 2007 Autophagy). The delivery of Atg8 to a functional (acidic) vacuole can be monitored by the detection of RFP and loss of GFP fluorescence.

Minor comments:

1) The authors place an emphasis on the negative curvature promoting lipids as crucial to their model but there are a couple of problems with this: First, another negative curvature promoting lipid (DAG) does not appear to promote the formation of the nuclear holes. This suggests that negative curvature promoting lipids are not equal in their ability to promote the formation of the nuclear holes and raises the possibility that PA is playing a specific role that should be more thoroughly considered. Second, as pointed out by the model in Figure 2B, negative curvature promoting lipids would need to be on the luminal leaflet to promote invagination of the INM. However, the NLS-PA sensor is detecting PA on the nucleoplasmic leaflet on the INM, which would promote evagination. Although, in the discussion, the authors argue against an invagination as a possible mechanism of vacuole internalization, the model in 2B and the emphasis on the negative curvature promoting property of the lipids is confusing and should be removed from the figure and title. There is also no data to show that there is a morphology analogous to that shown in Figure 2B.

2) The (de)convolution shown in Figure 2E is concerning. While some of the zeocin treated cells show a modest enrichment of the PA sensor at the nuclear rim in the raw images, the deconvolution seems to be removing too much of the signal. Even the untreated cells appear to have an enrichment of the PA sensor at the rim in the deconvolved image. I suggest a less stringent algorithm or confocal microscopy.

3) Do nuclear holes still form/contain vacuoles when the nucleus-vacuole junction is lost such as in *nvj1Δ/vac8Δ* cells? With this in mind, it would be informative to directly investigate if NVJ components are found at the NE-vacuole interface in this context.

4) Line 56 - suggest using the more common terminology of "nuclear envelope", which encompasses the multiple membranes that define the nuclear compartment as opposed to "nuclear membrane".

5) Line 241-243 - It would be helpful to see the data alluded to here.

Reviewer #3 (Comments to the Authors (Required)):

Summary:

This paper by Garcia et al. investigates structural changes of the yeast NVJ contact sites in response to addition of different genotoxins. They describe a new phenomenon whereby the vacuole becomes engulfed by the nucleus and propose this to be a mechanism for regulating autophagy by maximizing cargo interactions with non-sequestered "efficient" vacuoles. The authors claim that the enrichment of phosphatidic acid or free sterols is the main driver of this phenomenon.

Strengths:

The manuscript is easy to follow, and the data is well-presented. In general, the data presented fit the conclusions the authors

describe. The phenomenon observed is interesting although it remains unclear why certain genotoxins would drive this.

Weaknesses:

A major weakness of the paper is that it doesn't clearly define what nuclear holes are. The data presented is not sufficient to claim whether the holes are aberrations in the nuclear envelope or reorganization of the nucleoplasmic material. The study can be strengthened by more higher resolution imaging to clarify the identity of the holes.

Specific comments:

- It is unclear what the nuclear 'holes' morphology actually is. Since the majority of the study is dissecting this morphology, it is important that the authors clarify this. Are the holes due to reorganization of the internal nuclear features? Or modifications in the nuclear membrane itself? Using an ER marker (dsRed HDEL for example) will unambiguously answer this question as the Pus1 marker alone does not resolve that.
- How does the cell cycle profile look for cells treated with zeocin (rich media) vs zeocin (limiting media)? The authors showed the former (Fig 1C) but not the latter.
- How does carbon or nitrogen starvation affect the formation of the holes?
- Images corresponding to Fig 2A graphs should be shown in the Supp materials. Similarly for Fig 2G.
- As a control, how does the signal from the PA-sensor look in Pah1-KO and Dgk1-OE cells?
- How do Pah1-KO and Dgk1-OE strains respond to autophagy stimulation compared to Yeh2-KO?
- What is the role of the nER-vacuole junction in this process?

Reviewer #1 (Comments to the Authors (Required)):

Please find our answers preceded by the symbol ">" and in blue color font.

Please note that, in the main manuscript, all the new text appears in green font.

The study by Garcia et al is based on the observation that in the presence of reagents that lead to a G2/M arrest in budding yeast, there is an area in the nucleus that does not contain a nucleoplasmic marker (henceforth nuclear "hole", as the authors refer to it). The authors further show that these holes coincide with a dysfunctional vacuole. Using fluorescence imaging, the authors conclude that these holes are internalized vacuoles. If true, this is a remarkable observation, as an internalization of a vacuole into the nucleus has not been reported before, as far as this reviewer knows. Thus, the significance of this study rests on the quality of the data supporting this claim.

Previous work from several labs showed that during a G2/M cell cycle delay, the nucleus forms a loop-shaped protrusion (for example, Witkin et al 2012). Moreover, a recent study posted on BioRxiv from the Machin lab showed that this loop surrounds the vacuole. Thus, if the holes observed here are the same as the space associated with the nuclear envelope loops, the simplest explanation is that the holes are adjacent to the nucleus, or perhaps even engulfed by the nucleus, rather than internalized. The distinction between these two possibilities is straightforward: if the hole/vacuole is adjacent to the nucleus, the hole/vacuole lumen would be separated from the nucleoplasm by three membranes (the vacuolar membrane and the two nuclear membrane). If, however, the hole/vacuole is internalized, the nuclear membranes should be absent from the vacuole's periphery.

> we have now performed electron microscopy of both nocodazole-induced nuclear flares (as originally reported by Dr. Cohen-Fix and recently further developed by the team of Dr. Félix Machín, in Tenerife, Spain), presented in Fig 4C; and of *steΔ* cells treated with zeocin in order to induce formation of "our" nuclear holes, now presented in Fig 3C,D. The goal was to clarify the similarities / differences between both phenomena. The results conclusively show the different nature of both phenomena. Nocodazole triggers the expansion of the nuclear membrane, which starts curling around the vacuole, using it as a template, and the proximity of them gives rise to a "three membranes" profile (two membranes provided by the nuclear membrane and another one by the vacuole). The combination of the *steΔ* mutation and zeocin leads to the detection of structures fully within the nucleus, thus being surrounded by nuclear material from all sides and delimited by a double membrane, indistinguishable from the nuclear one. A schematic of the concerned structure is now drawn and explained in Fig 7B.

The present study has two major issues: first, the characterization of the spatial relationship between the hole/vacuole and nucleus is insufficient, and consequently the conclusion that the hole reflects vacuoles that are completely internalized by the nucleus, as the authors suggest, is premature. Second, the authors are presenting an endpoint condition. Live imaging would allow the determination of how the vacuole becomes surrounded/engulfed/internalized by the nucleus, and when during this process the vacuole becomes dysfunctional.

> We have now performed experiments in which Vph1-GFP was used to monitor the vacuoles and mCherry-Pus1 to monitor the nuclear holes (Fig 5C). Only exceptionally could we find a vacuole inside the hole. Further, we have used the Rosella construct (encoding a pH-stable RFP fluorochrome and a pH-sensitive GFP that allows to monitor for vacuolar degradative events, see details in Fig 6C legend, to score for functional / dysfunctional vacuoles, and have only found that, as a rule, the contents of the hole resemble those of the cytoplasm (Fig 6C), as estimated by their similar signal pixel intensity (Fig 6D). The notion that the content of the hole may mostly be of cytoplasmic origin is validated when observing the similar electron densities in the EM data presented in Fig 3C. Last, we also show now from z-stack analyses that the hole is a globular structure deeply embedded within the nucleus but that keeps a rope-like connection with the cytoplasm (Fig 3B). Combined with our previous MM4-64 (Fig 5B) and BCECF (Fig 5C) data, showing only eventual staining of vacuolar contents inside the hole, we conclude that the holes correspond to sequestration of cytoplasmic portions that become invaginated towards the nucleus interior and that, sporadically, can bring with them a “sequestered” vacuole. Yet, our revised message is that, in the most prevalent scenario, the holes do not contain vacuoles.

The comments listed below are aimed at addressed these and other issues. As it currently stands, the authors are grossly over-interpreting their data.

Major issues:

1. The main issues with this study are the lack of appropriate nuclear envelope and vacuole membrane markers and the absence of 3D reconstructions of both nuclei and vacuoles using, for example, confocal microscopy. This would enable the authors to distinguish between the two possibilities noted above (namely adjacent/engulfed vs internalized). Ideally this would be done by EM, but for an initial study fluorescence microscopy could suffice. It is quite possible that the vacuole is surrounded by the thin nuclear envelope protrusion (think shoelace around a ball) rather than surrounded by nuclear material from all sides (i.e. engulfed or internalized). This would also overcome the difficulties the authors have in observing the vacuole and will allow them to do live imaging so that they can follow the process by which the nucleus surrounds the vacuole as the cells become arrested in G2/M.

> As already described above, we have clarified the nature of the phenomenon by electron microscopy and additional strategies to monitor the vacuole, among which the use of Vph1-GFP mCherry-Pus1 time-lapse high-resolution TIRF microscopy to determine the kinetics of nuclear hole formation (Fig 5D). In addition, to meet the reviewer’s suggestion, we have repeated the experiment using a strain Nab2-GFP (to monitor the holes in the nucleoplasm with yet another marker) and Nup57-tDIMER (to define the nuclear periphery via a nucleoporin, as suggested). We have learnt from this additional experiment that:

- * the kinetics of nuclear hole ingression appear identical (the process takes from 15 to 30 min), irrespective of the marker used to monitor it.

- * the nuclear membrane invaginates in a manner that can concur with either nucleoporin splitting apart at the site of internalization, or with nucleoporins still present at the invaginating membrane (Fig 3A).

> Additionally, as already briefly mentioned in the previous point, we have performed z-stacks acquisition, deconvolution using Huygens and visualization using Imaris to obtain a glimpse of how holes may look in 3D. The data, part of which is now shown Fig 3B,

demonstrate that the holes are completely internalized in the nucleus, thus surrounded by nucleoplasmic material from all sides, as shown by the electron microscopy, yet are not sectioned from the cytoplasm, to which they remain connected by a rope-like structure.

2. Figure 2A-D is (a) not novel (see for example Han et al, JBC 2008 and other studies for the effect of DGK1 over expression and Pah1 and its various mutations on nuclear morphology)

> we are sorry, we have never intended to claim that the phenomenon of promoting negative curvature by Dgk1 overexpression is new. We only intended to use that exact knowledge to support the notion that our phenomenon could indeed be elicited through that excess.

and (b) over interpreted, because the authors do not know that PA of DAG are the causative agents of the nuclear hole- they can be metabolized into something else (for example, phospholipids) that is needed for membrane proliferation, for example.

> Since PA excess (as suggested by the phenotypes seen upon *DGK1* overexpression (Fig 2A) and *PAH1* deletion (Fig 2D), but not DAG (as suggested upon lack of phenotype during *PAH1-7A* overexpression, Fig 2A; or *DGK1* deletion; Fig 2D) matched an increase in nuclear hole detection, and precisely because the reviewer is right that this excess PA could be metabolized into something else, we used the nucleus-targeted PA-detecting biosensor to reinforce the notion that an increase of it could be observed, upon zeocin induction, at the inner nuclear membrane (Fig 2E). Yet, to strengthen the trust of the reviewer in the aforementioned PA-detecting strategy, we have included in this version controls in which a wide-cell PA sensor and the nucleus-targeted PA sensor are used in cells with excess PA, namely both *DGK1* overexpression and *PAH1* deletion (Fig S2B). This experiment shows that, during PA excess, the sensor strongly emits from the nuclear periphery, reinforcing our interpretation that Fig 2E is indicative of an increased detection of PA at the INM.

3. The authors conclude that PA accumulates in the inner nuclear membrane, in support of providing negative membrane curvature driving this membrane to bend in a concave fashion. However, their reporter (NLS-PA-sensor-mCherry) was directed to the nucleoplasm, which precluded them from examining whether PA is equally accumulating on the outer nuclear membrane, invalidating their model. This experiment should either be eliminated or carried out with the appropriate sensors to detect PA (or lack thereof) on the outer nuclear membrane.

> We agree with the reviewer that we should have assessed and included, in the original version, the pattern yielded by the PA-biosensor without NLS, in order to additionally see potential PA-enriched sites at the outer nuclear membrane. We yet disagree with the reviewer in the notion that “seeing PA-signals at the inner nuclear membrane invalidates the model”. In fact, in a phenomenon in which the nuclear membrane is to invaginate, negative curvature sites will have to be created BOTH at the INM and at the ONM. We have misled the reviewer by drawing in our previous version scheme only the points of the INM where this was expected. To catch up, in the revised version we have:

(a) updated the scheme presented in Fig 2B so that negative-curvature-requiring sites (exposed to the cytoplasm and the nucleoplasm) are clear both at the INM and at the ONM.

(b) performed the suggested experiment using a cell-wide-targeted PA-sensor (without NLS). We now see that the sensor congregates, in response to zeocin, either all over

the ONM or at various discrete spots on it (Fig S2C). This may represent sites at which PA gathers, perhaps contributing negative curvature potential.

4. To quantify the spatial relationship between the holes/vacuoles and the nucleolus, the authors should repeat Figure 3 using a fluorescence vacuolar membrane marker (e.g. Vph1-GFP) in addition to the nucleolar marker (and preferably also a nuclear marker) and carry out 3D reconstructions based on confocal images. This is an important result because if the holes are not near the nucleolus, that will distinguish them from the mitotic nuclear membrane extensions described previously. Note, however, that yeast may have multiple nuclei and there are likely many vacuoles that are not associated with the nucleus anyway. When conducting these experiments, it will be beneficial to account for all cellular vacuoles.

> We have already clarified above that the holes and the “mitotic nuclear extensions” are of a different nature. Yet, this did not rule out the possibility that the membrane surrounding the nucleolus is the one supporting hole ingress. We have now used Net1-GFP as a marker of the rDNA, to define the inner core of the nucleolus, and mCherry-Pus1 to locate the formation and position of holes. We have then performed time-lapse microscopy to monitor the link between the beginning of hole internalization and the position of the rDNA at each of those moments. We now define that the hole enters the nucleus almost always from this precise location (Fig 4A). As such, the old quantification data, showing that the holes do not always coincide with the nucleolus when studied at steady state, are exploited in the revised version to illustrate that, once inside, the internalized globular structure has a “certain mobility”, and can “dive away” from the initial entry nuclear domain.

5. What fraction of the total vacuolar volume is present in the nuclear hole? According to Figure 3E, where the authors used a vacuolar lumen marker, only a tiny fraction of the total vacuolar mass is nuclear-associated. If this is the case, it's not clear how the dysfunction of this small portion has such a large effect on autophagy (Figure 4).

> In view of our revised work, this concern does no longer apply. Thank you.

Is it possible that the effect of genotoxic agents on autophagy is a general one, not related to the specific nuclear-associated vacuole?

> A general yet very mild stimulation of autophagy has been reported for multiple genotoxins and even for direct cutting of the DNA in the past (Eapen...Haber 2017 PNAS). Yet, as we mention, the “hole” is not a common feature triggered by other genotoxins.

Also, the authors state "a striking correlation between nuclear hole presence and autophagic flux. However, they did not examine nuclear holes in the presence of rapamycin, which is the condition under which autophagy was examined. Rapamycin is known to inhibit phospholipid synthesis and it is quite possible that under these conditions the number of holes is significantly diminished.

> this is a fundamental and very important concern that we had neglected! We have now repeated the kinetic experiments in which we add zeocin, in order to promote hole formation, and assessed the actual presence of nuclear holes if we add or not rapamycin on top. Not only do we observe that the presence of zeocin alone for longer times (up to 19 h) than assessed in the past gives rise to a permanently increasing number of cells in the population bearing at least one nuclear hole, but, crucial to support the model, the addition of rapamycin does not alter in the least this number. This new piece of information has been included in Fig S4B.

6. It is interesting that the vacuoles that are associated with the nucleus are less functional, but unless the authors can demonstrate unequivocally that the holes/vacuoles are inside the nucleus, the section on "vacuole sequestration" is grossly over-interpreted and should be removed or re-written (including Figure 5).

> As already evoked, we have now performed experiments in which Vph1-GFP was used to monitor the vacuoles and mCherry-Pus1 to monitor the nuclear holes (Fig 5D). Only exceptionally could we find a vacuole inside the hole. Further, we have used the Rosella construct (encoding a pH-insensitive RFP fluorochrome and a pH-sensitive GFP that allow to monitor for vacuolar degradative events, see details in Fig 6C legend) to score for functional / dysfunctional vacuoles, and have only found that, as a general rule, the contents of the hole resemble those of the cytoplasm (Fig 6D,E). Combined with our previous MM4-64 (Fig 5B) and BCEFC (Fig 5C) data, we conclude that the sequestration of cytoplasmic portions sporadically come with a "sequestered" vacuole, but that the most prevalent scenario is that in which the hole does not contain vacuoles.

> In view of this change in our initial vision of the phenomenon, and to try to understand how the ingress of cytoplasmic contents within the nucleus could eventually positively impact autophagy, our time-lapse analysis of Vph1-GFP mCherry-Pus1 made us realize that the formation of the hole systematically precedes the docking and fusion of small Vph1-positive vacuoles or vesicles with the main vacuole. This event happened to be highly statistically significant (Fig S3C). Combined with the observation that the nuclear hole recurrently invaginates from the part of the nuclear membrane in close contact with the vacuole, that is, the one surrounding the nucleolus (Fig 4A), we now propose that the process of cytoplasm ingress into the nucleus may create a suction effect locally that accelerates the docking of Vph1-transporting vesicles whose destination is the vacuole. Vph1 being the proton pump key to confer acidity (thus autophagic ability) to the vacuole, we propose this may contribute to the increase in autophagic capacity. This "enhanced docking" model that would underlie a stimulation of autophagy is equally valid for autophagic cargoes. This has now been included as a revisited model in Fig 7A, and appropriately discussed.

Minor issues:

1. The title of the manuscript is an over/mis-interpretation of the data does not reflect the novel findings in the study.

> We hope that, upon reading the revised version, the updated title seems appropriate and fair to this reviewer.

2. Please provide representative images to go with Figure 2A so that the reader can evaluate the morphology of the nuclei under the various conditions.

> Done, now presented in new Fig S2A.

3. It is not clear what the authors meant by "embedded sterols can act like adjuvants".

> We intended to say that the presence of free sterols embedded in the membranes could foster, contribute to further boosting, the phenomenon of nuclear hole formation. This sentence has been re-written to avoid any misunderstanding.

4. Figure 3D: please use fluorescently tagged nuclear envelope markers (NPC subunits, LINC complex, Heh1, etc) rather than an ER marker in conjunction with a fluorescently tagged

vacuolar membrane marker (e.g. VPH1) and image the cells using confocal/high-resolution microscopy.

> As suggested, not only in that figure but in several of the new figures, we have removed the ER marker Sec63, used the nucleoporin Nup57 tagged with tDIMER, and used Vph1-GFP, both imaged by high-resolution TIRF microscopy.

We thank this reviewer for his/her time and very pertinent and helpful suggestions.

Reviewer #2 (Comments to the Authors (Required)):

Please find our answers preceded by the symbol ">" and in blue color font.

Please note that, in the main manuscript, all the new text appears in green font.

Review comment:

This paper by Garcia et al. explores an unusual phenotype in budding yeast where there is a "hole" in the nucleus that is visualized by light microscopy using a nucleoplasmic reporter following treatment of cells with genotoxins. The authors make the provocative conclusion that this hole represents a vacuole that has been fully internalized into the nucleus. The authors demonstrate that the formation of the nuclear hole requires phosphatidic acid (PA) through several different genetic perturbations and by directly visualizing PA at the INM using a genetically encoded fluorescent PA sensor. They further provide evidence that such vacuole internalization might limit autophagy flux. Overall, the data are of good quality and investigate a unique and potentially interesting phenomenon that has not been documented in the past. Thus, there is the potential that new biology will stem from these foundational observations. Were the data to stand solely on their characterization and preliminary functional analysis of this phenotype, I would be supportive of publication in its current form, however, the authors choose a very provocative conclusion (that the vacuole is somehow internalized into the nucleus), which is not supported by the data and would require substantial experiments to solidify.

Major Point:

1) Although the authors seemingly observe vacuoles in the nucleus, this conclusion is reliant on wide-field fluorescence microscopy, which has insufficient resolution to rule out that vacuoles might simply be surrounded by a nuclear envelope lobulation instead of being fully internalized. The suggestion is to substantially tone down the conclusions of the manuscript in this respect and to consider more likely explanations (see point 2, below). The title would also need to be changed. If the authors choose to be more definitive experimentally, they will need substantial high quality ultrastructural data that would really need to be serial section EM or FIB-SEM in order to capture the entire nuclear contour. In addition, or alternatively, they could provide direct evidence that the internalized vacuole is biochemically segregated from the cytoplasm.

> we have now performed electron microscopy of *steΔ* cells treated with zeocin in order to induce formation of "our" nuclear holes, now presented in Fig 3C,D. This led to the detection of structures fully within the nucleus, as such being surrounded by nuclear material from all sides, and delimited by a double membrane indistinguishable from the nuclear one. A schematic of the concerned structure is now drawn and explained in Fig 7B.

> we have performed z-stacks acquisition, deconvolution using Huygens and visualization using Imaris to obtain a glimpse of how holes may look in 3D. The data, part of which is now shown in Fig 3B, demonstrate that the holes are completely internalized in the nucleus, thus surrounded by nucleoplasmic material from all sides, in agreement with the electron microscopy, yet are not sectioned from the cytoplasm, to which they remain connected by a rope-like structure.

> we have now performed time-lapse experiments in which Vph1-GFP was used to monitor the vacuoles and mCherry-Pus1 to monitor the nuclear holes (Fig 5D). Only exceptionally could we find a vacuole inside the hole. Further, we have used the Rosella

construct suggested by this reviewer (encoding a pH-insensitive RFP fluorochrome and a pH-sensitive GFP that allow to monitor for vacuolar degradative events, see details in Fig 6C legend) to score for functional / dysfunctional vacuoles, and have only found that, as a rule, the contents of the hole resemble those of the cytoplasm (Fig 6C), as estimated by their similar signal pixel intensity (Fig 6D). The notion that the content of the hole may mostly be of cytoplasmic origin is validated when observing the similar electron densities in the EM data presented in Fig 3C. Combined with our previous MM4-64 (Fig 5B) and BCECF (Fig 5C) data, both showing sporadic dye of vacuolar contents within the nucleus, we conclude that the holes correspond to sequestration of cytoplasmic portions that become invaginated towards the nucleus interior and that, sporadically, can bring with them a “sequestered” vacuole. Yet, our revised message is that, in the most prevalent scenario, the hole does not contain vacuoles.

> we have consequently adapted the title of our manuscript, which we hope is now convenient and fair for this reviewer.

Additional Points for authors/editors to consider to strengthen the manuscript:

2) Given the requirement for PA and the proximity of the nuclear holes to the nucleolus (which is established to be prone to membrane expansion upon perturbation to PA metabolism), the authors should consider the possibility that the hole they see results from an extended nuclear flare as defined by work from Orna Cohen-Fix.

> we have now performed electron microscopy of both nocodazole-induced nuclear flares (as originally reported by Dr. Cohen-Fix and recently further developed by the team of Dr. Félix Machín, in Tenerife, Spain), presented in Fig 4C, and of *steΔ* cells treated with zeocin in order to induce formation of “our” nuclear holes, now presented in Fig 3C,D. The results conclusively show the different nature of both phenomena. Nocodazole triggers the expansion of the nuclear membrane, which starts wrapping the vacuole, using it as a template, and the proximity of them gives rise to a “three membranes” profile (two membranes provided by the nuclear membrane and another one by the vacuole). Yet, as mentioned before, the combination of the *steΔ* mutation and zeocin leads to the detection of structures fully within the nucleus. These structures are therefore defined by a double membrane, indistinguishable from the nuclear one. As such, we reliably conclude that the phenomenon we describe is of a different nature as that described by Dr. Cohen-Fix.

A more careful quantification of the proximity of the “hole” to the nucleolus would seem warranted. The authors could also consider performing time-lapse microscopy following zeocin treatment in the context of NE, nucleolar and vacuolar labels to understand the biogenesis of these structures more fully.

> We have, as suggested, used Net1-GFP as a marker of the rDNA, to define the inner core of the nucleolus, and mCherry-Pus1 to locate the formation and position of holes. We have then performed time-lapse microscopy to monitor the link between the beginning of hole ingression and the position of the rDNA at each of those moments. The “careful” study, as suggested by the reviewer, has proven very pertinent, as we now define that the hole ingresses in the nucleus almost always from this precise location. As such, the old quantification data, saying that the holes do not always coincide with the nucleolus when studied at steady state, are exploited in the revised version to illustrate that, once inside, the internalized globular structure has a “certain mobility”, and can “dive away” from the initial

entry

nuclear

domain.

3) The authors' conclusion that the increase in autophagic flux following zeocin treatment is due to the internalization of low autophagic potential vacuoles rather than a general increase in autophagy caused by the zeocin itself needs to be tested more rigorously. The idea that vacuoles are not identical in their potential to degrade autophagic cargo is interesting and testable. As a tool, I suggest looking at Atg8 tagged with a tandem GFP-RFP tag utilizing a pH sensitive GFP (e.g. the Rosella tag, Rosado et al. 2007 Autophagy). The delivery of Atg8 to a functional (acidic) vacuole can be monitored by the detection of RFP and loss of GFP fluorescence.

> Thank you for having suggested this very useful and elegant tool. We have tested it in our system. Given the current, revised message that the content of the "holes" corresponds to engulfed cytoplasmic material, the tool has allowed us to:

- * check that *bona-fide* vacuoles are functional (RFP detection and GFP signal loss) when exposing cells to zeocin + rapamycin (in opposition to the control, in which rapamycin-exposed cells previously exposed to the protonophore CCCP become dysfunctional).

- * highlight by fluorescence microscopy in a yet fourth manner (in addition to mCherry-Pus1, Nab2-GFP and DAPI staining) that the nuclear holes content is different from the rest of the nucleoplasm. Indeed, since the GFP and RFP signals are tightly packed inside the nucleus, the same signals appear much lighter from the hole (Fig 6C,D).

- * show that there is no evidence for degradative events inside the "nuclear holes", since GFP signal is never absent when RFP is present.

- * reinforce the notion that the main content of the holes may be of cytoplasmic origin, as the measured pixel intensity is similar in both compartments (Fig 6D).

Minor comments:

1) The authors place an emphasis on the negative curvature promoting lipids as crucial to their model but there are a couple of problems with this: First, another negative curvature promoting lipid (DAG) does not appear to promote the formation of the nuclear holes. This suggests that negative curvature promoting lipids are not equal in their ability to promote the formation of the nuclear holes and raises the possibility that PA is playing a specific role that should be more thoroughly considered.

> Although this is a fair criticism, it could happen that some but not all the lipids capable of conferring a feature to a membrane are at play when it comes to a given location. Of particular relevance to our context, we now define better that the nuclear membrane subdomain mainly involved in invaginating towards the nucleus to give rise to the holes is the one surrounding the nucleolus (Fig 4A). The team of Dr. Symeon Siniossoglou recently defined that this is the very specific place at which the enzyme Lro1 esterifies DAG towards TAG, and the regulation of this reaction is key to permit membrane remodeling events (Barbosa et al 2019 Dev. Cell). Thus, it could happen that DAG is rapidly processed, even if overexpressed, in response to the cue that triggers nuclear holes formation. The mention to this specific regulation has now been included in the discussion.

Second, as pointed out by the model in Figure 2B, negative curvature promoting lipids would need to be on the luminal leaflet to promote invagination of the INM. However, the NLS-PA sensor is detecting PA on the nucleoplasmic leaflet on the INM, which would promote evagination. Although, in the discussion, the authors argue against an invagination as a possible mechanism of vacuole internalization, the model in 2B and the emphasis on the negative curvature promoting property of the lipids is confusing and should be removed from the figure and title. There is also no data to show that there is a morphology analogous to that shown in Figure 2B.

> We agree with the reviewer that the explanation was misleading, and the scheme, incomplete. It is also true that, for inherent technical limitations, we cannot assess the PA pool on the luminal side of both the inner and the outer nuclear membranes. But it was possible for us to complete our analysis by studying the cytoplasmic leaflet of the outer nuclear membrane (ONM). In fact, in a phenomenon in which the nuclear membrane is to invaginate, negative curvature sites will have to be created, which can be monitored, BOTH at the nucleoplasmic leaflet of the INM and at the cytoplasmic leaflet of the ONM. In this revised version, we have:

(a) updated the scheme presented in Fig 2B so that negative-curvature-requiring sites are clear both at the INM and a the ONM.

(b) added an experiment using a PA-sensor without NLS. We now see that the sensor congregates, in response to zeocin, at various discrete spots at the ONM. This may represent sites at which PA congregates, perhaps contributing negative curvature potential. These data are now included in Figure S2C and described in the results section.

2) The (de)convolution shown in Figure 2E is concerning. While some of the zeocin treated cells show a modest enrichment of the PA sensor at the nuclear rim in the raw images, the deconvolution seems to be removing too much of the signal. Even the untreated cells appear to have an enrichment of the PA sensor at the rim in the deconvolved image. I suggest a less stringent algorithm or confocal microscopy.

> We are very sorry that the image yielded a result that makes the phenotype appear as too "strong". The concern is that, if we are to lower down the stringency of the (de)convolution algorithm, the loss of quality during image printing may end up with the opposite effect: a lack of phenotype visualization. Thus, we humbly post-pone this request to the arbitration by the editorial team in case of acceptance.

3) Do nuclear holes still form/contain vacuoles when the nucleus-vacuole junction is lost such as in *nvj1Δ/vac8Δ* cells? With this in mind, it would be informative to directly investigate if NVJ components are found at the NE-vacuole interface in this context.

> This is a very interesting projection for our work and we are very keen on investigating it. Yet, with all the due respect, we dare answer that this would represent a full, additional characterization and, as such, needs to be implemented as a follow-up of the current study.

4) Line 56 - suggest using the more common terminology of "nuclear envelope", which encompasses the multiple membranes that define the nuclear compartment as opposed to "nuclear membrane".

> changed as suggested. Thank you.

5) Line 241-243 - It would be helpful to see the data alluded to here.

> The reviewer was meaning here the data in which the nucleus-targeted PA sensor is monitored in the WT strain in response to zeocin yet in minimal medium. These data are now included in Fig S2D.

We want to take the opportunity to sincerely thank this reviewer for his/her helpful and pertinent assessment, which has truly helped improve the manuscript.

Reviewer #3 (Comments to the Authors (Required)):

Please find our answers preceded by the symbol ">" and in blue color font.

Please note that, in the main manuscript, all the new text appears in green font.

Summary:

This paper by Garcia et al. investigates structural changes of the yeast NVJ contact sites in response to addition of different genotoxins. They describe a new phenomenon whereby the vacuole becomes engulfed by the nucleus and propose this to be a mechanism for regulating autophagy by maximizing cargo interactions with non-sequestered "efficient" vacuoles. The authors claim that the enrichment of phosphatidic acid or free sterols is the main driver of this phenomenon.

Strengths:

The manuscript is easy to follow, and the data is well-presented. In general, the data presented fit the conclusions the authors describe. The phenomenon observed is interesting although it remains unclear why certain genotoxins would drive this.

> Thank you for your positive perception. We do not know yet why the concerned genotoxins elicit the phenomenon, but we purport that they trigger a given set of alterations in the metabolism of lipids that are supportive to the development of this phenomenon. We have better developed this idea in the discussion.

Weaknesses:

A major weakness of the paper is that it doesn't clearly define what nuclear holes are. The data presented is not sufficient to claim whether the holes are aberrations in the nuclear envelope or reorganization of the nucleoplasmic material. The study can be strengthened by more higher resolution imaging to clarify the identity of the holes.

> we have now performed electron microscopy of *steΔ* cells treated with zeocin in order to induce formation of the nuclear holes, now presented in Fig 3C,D. This led to the detection of structures fully within the nucleus, and as such being surrounded by nuclear material from all sides and delimited by a double membrane indistinguishable from the nuclear one. A schematic of the concerned structure is now drawn and explained in Fig 7B.

> we have performed z-stacks acquisitions, deconvolution using Huygens and visualization using Imaris to obtain a glimpse of how holes may look in 3D. The data, part of which is now shown in Fig 3B, demonstrate that the holes are completely internalized in the nucleus, thus surrounded by nucleoplasmic material from all sides, in agreement with the electron microscopy, yet are not sectioned from the cytoplasm, to which they remain connected by a rope-like structure.

> we have now performed time-lapse experiments in which Vph1-GFP was used to monitor the vacuoles and mCherry-Pus1 to monitor the nuclear holes (Fig 5D). Only exceptionally could we find a vacuole inside the hole. Further, we have used the Rosella construct (encoding a pH-insensitive RFP fluorochrome and a pH-sensitive GFP that allow to monitor for vacuolar degradative events, see details in Fig 6C legend) to score for functional / dysfunctional vacuoles, and have only found that, as a rule, the contents of the hole resemble those of the cytoplasm (Fig 6C), as estimated by their similar signal pixel intensity (Fig 6D). The notion that the content of the hole may mostly be of cytoplasmic origin is further validated when observing the similar electron densities in the EM data presented in Fig 3C.

Combined with our previous MM4-64 (Fig 5B) and BCECF (Fig 5C) data, showing only eventual staining of vacuolar contents inside the hole, we conclude that the holes correspond to sequestration of cytoplasmic portions that become invaginated towards the nucleus interior and that, sporadically, can bring with them a “trapped” vacuole. Thus, our revised message is that, in the most prevalent scenario, the hole does not contain vacuoles.

> we have consequently adapted the title of our manuscript, which we hope is now convenient and fair for this reviewer.

Specific comments:

- It is unclear what the nuclear 'holes' morphology actually is. Since the majority of the study is dissecting this morphology, it is important that the authors clarify this. Are the holes due to reorganization of the internal nuclear features?

> we hope that this specific concern is now satisfied by the experiments we have just described above.

Or modifications in the nuclear membrane itself? Using an ER marker (dsRed HDEL for example) will unambiguously answer this question as the Pus1 marker alone does not resolve that.

> As already described above, we have clarified the nature of the phenomenon by electron microscopy and additional strategies to monitor the vacuole, among which the use of Vph1-GFP mCherry-Pus1 time-lapse high-resolution TIRF microscopy to determine the kinetics of nuclear hole formation (Fig 5D). In addition, to meet the reviewer's concern regarding the nuclear membrane, we have repeated the experiment using a strain Nab2-GFP (to monitor the holes in the nucleoplasm with yet another marker) and Nup57-tDIMER (to define the nuclear periphery via a nucleoporin). We have learnt from this additional experiment that:

* the kinetics of nuclear hole ingression appear identical (the process takes from 15 to 30 min), irrespective of the marker used to monitor it.

* the nuclear membrane invaginates in a manner that can concur with either nucleoporin splitting apart at the site of ingression, or with nucleoporins still present at the invaginating membrane (Fig 3A).

- How does the cell cycle profile look for cells treated with zeocin (rich media) vs zeocin (limiting media)? The authors showed the former (Fig 1C) but not the latter.

> this information has now been included in Fig S1C.

- Images corresponding to Fig 2A graphs should be shown in the Supp materials. Similarly for Fig 2G.

> this information has been included now in Fig S2A and in Fig S2E.

- As a control, how does the signal from the PA-sensor look in Pah1-KO and Dgk1-OE cells?

> this pertinent control has now been included in Fig S2B. It shows, in agreement with our interpretation of the experiment shown in Fig 2E, that an excess of PA triggers the accumulation of the sensor at the nuclear membrane.

- How do Pah1-KO and Dgk1-OE strains respond to autophagy stimulation compared to Yeh2-KO?

> The phenotype of Pah1-KO with respect to autophagy has been already studied in its genetic mimic *nem1Δ* by the team of Dr. Ushimaru. They report that the autophagy is severely hampered. As discussed in the previous version, and maintained in this one, the process of hole formation (nuclear membrane ingression) is so dramatic in *pah1Δ*, *nem1Δ* or Dgk1^{OE} cells, that it leads to a *bona-fide* internalization of the vacuole, which may indeed hamper its interaction with autophagic cargoes.

- What is the role of the nER-vacuole junction in this process?

- How does carbon or nitrogen starvation affect the formation of the holes?

> we take note of both these relevant questions and are keen on investigating them. Yet, we want to respectfully answer that this may constitute the expansion of the current work towards the characterization of different metabolic scenarios and of additional determinants. As such, we humbly think this belongs to a follow-up of the current study.

We want to thank this reviewer for his/her insightful and good-willing comments and suggestions.

February 18, 2022

Re: Life Science Alliance manuscript #LSA-2021-01160-TR-A

Dr. María Moriel-Carretero
Centre de Recherche en Biologie cellulaire de Montpellier
(CRBM)
Université de Montpellier,
Centre National de la Recherche Scientifique
1919 Route de Mende
Montpellier 34293
France

Dear Dr. Moriel-Carretero,

Thank you for submitting your revised manuscript entitled "Nuclear ingression of cytoplasmic bodies accompanies a boost in autophagy" to Life Science Alliance. The manuscript has been seen by the original reviewers whose comments are appended below. While the reviewers continue to be overall positive about the work in terms of its suitability for Life Science Alliance, some important issues remain.

Overall, the reviewers are now more positive and feel that the paper has much improved. A major concern raised by Rev1 is that a critical experiment previously suggested by Rev 2, namely determining the presence of nuclear holes when the nuclear-vacuolar junction (NVJ) is disrupted, was not performed. This is especially important as the authors want to argue a causative link between hole formation and autophagic flux. In the same line also, Rev 3 says that remains unclear under what physiological conditions this happens, what is the physiological trigger, and what the role of NVJ is. No experiments were done to determine whether this is related to the NVJ or can happen in absence of it. Thus, this key experiment needs to be performed in your revised version before resubmission. All the other concerns raised by the reviewers should be addressed as well. We, thus, encourage you to submit a revised version of the manuscript back to LSA that responds to all the reviewers' points.

Our general policy is that papers are considered through only one revision cycle; however, given that the suggested changes are relatively minor, we are open to one additional short round of revision. Please note that I will expect to make a final decision without additional reviewer input upon resubmission.

Please submit the final revision within two to three months, along with a letter that includes a point by point response to the remaining reviewer comments.

To upload the revised version of your manuscript, please log in to your account: <https://lsa.msubmit.net/cgi-bin/main.plex>
You will be guided to complete the submission of your revised manuscript and to fill in all necessary information.

B. MANUSCRIPT ORGANIZATION AND FORMATTING:

Sincerely,

Novella Guidi, PhD
Scientific Editor

Reviewer #1 (Comments to the Authors (Required)):

In their revised version, Garcia et al provide convincing evidence that in the presence of Zeocin, or elevated phosphatidic acid levels, the nucleus contains structures that lack nucleoplasmic content. While in the original version of this manuscript the author suggested that these nuclear "holes" contain vacuoles, upon further inspection, and in response to the reviewers' comments, the authors now conclude that these holes are not of vacuolar origin.

Overall, the manuscript is much improved, with time courses depicting the formation of nuclear holes and EM images that are suggestive of their origins and structure (namely, not vacuolar). However, given the emphasis the authors put on the relationship between the holes and the nuclear-vacuolar junction (NVJ), a critical experiment suggested by another reviewer, namely determining the presence of nuclear holes when the NVJ is disrupted, was not performed. This is especially important as the authors want to argue a causative link between hole formation and autophagic flux. This could be tested by disrupting the NVJ, assuming that NVJs are important for hole formation (and if they are not, that is something the reader should definitely know). The response to zeocin could have led to nuclear holes and changes in vacuole activity via two independent pathways. In fact, it's not even clear what is special about zeocin that leads to nuclear hole formation, when other treatments that either cause DNA damage or lead to a G2/M arrest, do not. As it stands, the simplest explanation for these intranuclear invaginations is that the extended G2/M delay causes membrane expansion, but unlike the previously described mitotic flares, for a yet unknown reason these protrusions extend towards the nuclear interior.

Minor comments:

1. The authors show that the presence of nuclear holes is increased in the presence of zeocin, a DNA damaging agent, and phosphatidic acid, a precursor of phospholipid synthesis and also promoted negative membrane curvature. The authors focus on the latter, but work from the Carmen Lab has shown that *pah1Δ* cells, for example, have increased level of phospholipids, which could be the underlying cause of the nuclear holes. This would be consistent with the observation that DAG does not promote hole formation, despite promoting negative membrane curvature. Thus, the authors should tone their conclusion that PA promoted holes by promoting negative membrane curvature.
2. I agree with one of the other reviewers that the deconvolution in Figure 2E is extreme. The authors must discuss in the body of the manuscript how they got to this result. It might also help to apply the same deconvolution parameters to a protein that is not expected to be at the nuclear periphery (e.g. Pus1-mCherry). Also, rather than untreated, the authors should use HU treated cells, which are also arrested (so have equally large nuclei) but do not show holes. Finally, n=7 is not sufficient. It should be no problem to image and quantify dozens of cells.
3. The Summary blurb is inappropriate because the authors did not demonstrate that the hole are important for other organelles or for metabolic adaptation.

Reviewer #2 (Comments to the Authors (Required)):

Garcia et al have added substantial new data to their manuscript revision. The emerging picture that the nuclear holes are not really associated with vacuoles and might be capturing cytosol instead is interesting but raises the inevitable possibility that zeocin treatment leads to innumerable changes to cellular physiology and an expansion of the nuclear envelope, which lobulates giving the impression of a selective capture event. I still think that their thorough exploration of this phenomenon should be published but there remains a tendency to overreach with their conclusions. Any notion that there is a "suction" of the cytosol, for example, needs to be clearly defined as being speculative. Otherwise, please address the following remaining points:

- 1) Throughout manuscript - please use the term "nuclear envelope" instead of "nuclear membrane" as there are multiple nuclear membranes.
- 2) There is still insufficient evidence to suggest that the increase in autophagic flux is somehow due to the capture of cytosol within the nucleus. Another explanation would be that the zeocin elicits an autophagic response that is enhanced by the rapamycin treatment. This appears to be the case in Fig. 6A where there is a clear increase in autophagic flux following zeocin but prior to rapamycin treatment. This should be stated somewhere in the text as an alternative explanation.
- 3) Please remove the fluorescence microscopy insets from Fig. 3C as it gives the misleading impression that the authors performed correlative light and electron microscopy on those cells.
- 4) Line 186 - As the PA sensor is not completely nuclear (i.e. there remains a cytosolic pool), it cannot be interpreted that there

is a localization specifically at the ONM. Suggest to use "nuclear envelope" instead.

Reviewer #3 (Comments to the Authors (Required)):

This paper by Garcia et al describes an interesting phenomenon that happens in yeast cells whereby the nucleus engulfs portion of the cytoplasm. They associate the phenomenon with changes in autophagic flux.

It remains unclear under what physiological conditions this happens, what is the physiological trigger, and what the role of NVJ is. The authors start by talking about contact sites which sets the stage for the reader to expect more about the role of the NVJ contact site in this phenomenon, and yet no experiments were done to determine whether this is related to the NVJ or can happen in absence of it.

That being said, major issues raised during the first revision round regarding the nature of the "holes" in the nucleus were addressed by performing high resolution confocal imaging. Note that transmission electron microscopy is not sufficient to comment on the 3D arrangement of the holes because this technique also images a section across the cell. The authors have made the revisions necessary to publish this paper in the current revised format. This paper, although descriptive and as the author themselves mention, warrants further analysis, will be of interest to the community and the tools used to do the analysis will be of interest too.

In conclusion, I support publication in the current format.

In this second round of revision, we have fully addressed the reviewers' new concerns. Our answers to them can be found preceded by the symbol ">" and in blue font. Please note that, within the main manuscript body, new text changes can be identified in pink font.

Thank you.

Reviewer #1 (Comments to the Authors (Required)):

1. However, given the emphasis the authors put on the relationship between the holes and the nuclear-vacuolar junction (NVJ), a critical experiment suggested by another reviewer, namely determining the presence of nuclear holes when the NVJ is disrupted, was not performed. This is especially important as the authors want to argue a causative link between hole formation and autophagic flux. This could be tested by disrupting the NVJ, assuming that NVJs are important for hole formation (and if they are not, that is something the reader should definitely know).

- The hole formation stems from the nuclear envelope subdomain that surrounds the nucleolus. This same domain is engaged in a tight and dynamic contact with the vacuole, the nucleus-vacuole junction (NVJ). This participation in two different processes raises the possibility that disrupting the NVJ may affect hole formation. The requested experiment is thus to delete *NVJ1* and to assess the impact this may have on the formation of nuclear holes. We have now completed this experiment and observed that, in response to zeocin, cells form nuclear holes more readily in the absence of *Nvj1* (New Fig 5A, compare fold-change from 0 to 120 min treatment in WT versus *nvj1Δ*). Yet, the final hole formation values are unchanged (Fig 5A, time 210 min). We have also performed the experiment in a *yeh2Δ* strain, where the basal level of holes and the readiness to form them upon zeocin treatment is increased. In this case, the absence of *Nvj1* does not strengthen the phenotype further (Fig 5A, *yeh2Δ* versus *yeh2Δ nvj1Δ*). Together, these data suggest that the nuclear envelope subdomain that surrounds the nucleolus is indeed shared between its contact with the vacuole and its availability to engage in nuclear hole formation.

2. The authors show that the presence of nuclear holes is increased in the presence of zeocin, a DNA damaging agent, and phosphatidic acid, a precursor of phospholipid synthesis and also promoted negative membrane curvature. The authors focus on the latter, but work from the Carmen Lab has shown that *pah1Δ* cells, for example, have increased level of phospholipids, which could be the underlying cause of the nuclear holes. This would be consistent with the observation that DAG does not promote hole formation, despite promoting negative membrane curvature. Thus, the authors should tone their conclusion that PA promoted holes by promoting negative membrane curvature.

- This possibility has now been openly explained in lines 170-171.

3. I agree with one of the other reviewers that the deconvolution in Figure 2E is extreme. The authors must discuss in the body of the manuscript how they got to this result. It might also help to apply the same deconvolution parameters to a protein that is not expected to be at the nuclear periphery (e.g. Pus1-mCherry). Also, rather than untreated, the authors should use HU treated cells, which are also arrested (so have equally large nuclei) but do not show holes. Finally, n=7 is not sufficient. It should be no problem to image and quantify dozens of cells.

- Our intention when using the convolution/deconvolution of the images was to help the reader appreciate the enrichment in sensor signal at the membrane. We never intended to make of this a "method" whose use requests further and further validation. Given the fact that the use of this tool has raised the concern in two reviewers, we have decided that it is best NOT TO USE it, and to present exclusively the raw, unprocessed signals. Since the phenotype is nevertheless

seen this way, it is simpler, neater, not controversial and, overall, does not need any additional controls as the ones suggested. We now present 8 examples of the untreated condition (ie, the typical, nucleoplasmic-wide distribution) and 16 examples of the zeocin condition where the sensor becomes enriched at the membrane.

- We drew a line through each nucleus presenting the concerned phenotypes, that is, homogeneous nucleoplasmic signals or PA sensor signals enriched at the nuclear membrane. We then plotted the pixel intensity along that line. We did this for n=7 of each type and presented a graph. The goal of such graph was not to do any statistics, but to illustrate (using the raw images as the source for the quantification) that there was an objective (Image J-validated) enrichment at the periphery in the +zeocin condition, irrespective of our human, visual perception of the images. The reviewer argued, nevertheless, that n=7 was not enough. We have now increased up to 27 events for the nucleoplasmic signals and 44 events for the membrane-enriched ones. The overall graph has not changed. We expect this analysis is considered as more robust now, thank you.

4. The Summary blurb is inappropriate because the authors did not demonstrate that the hole are important for other organelles or for metabolic adaptation.

- We have now revised and adapted our summary blurb.

Reviewer #2 (Comments to the Authors (Required)):

please address the following remaining points:

1) Throughout manuscript - please use the term "nuclear envelope" instead of "nuclear membrane" as there are multiple nuclear membranes.

- We have changed the term accordingly in most places (identifiable by their pink font). Whenever it still reads "nuclear membrane", it is because it is pertinent this way.

2) There is still insufficient evidence to suggest that the increase in autophagic flux is somehow due to the capture of cytosol within the nucleus. Another explanation would be that the zeocin elicits an autophagic response that is enhanced by the rapamycin treatment. This appears to be the case in Fig. 6A where there is a clear increase in autophagic flux following zeocin but prior to rapamycin treatment. This should be stated somewhere in the text as an alternative explanation.

- This explanation has now been included in line 322-324.

3) Please remove the fluorescence microscopy insets from Fig. 3C as it gives the misleading impression that the authors performed correlative light and electron microscopy on those cells.

- Insets have now been removed.

4) Line 186 - As the PA sensor is not completely nuclear (i.e. there remains a cytosolic pool), it cannot be interpreted that there is a localization specifically at the ONM. Suggest to use "nuclear envelope" instead.

- This change has been included by removing "the ONM" from the sentence. This way, the old sentence read "suggesting concentration or at least exposure of PA

- at those sites in the ONM”, and now it reads “suggesting concentration or at least exposure of PA at those sites”.
- For the same reason, we have also substituted “the ONM” by “the nuclear envelope”, as suggested, in line 175 (indicated in pink font).

Reviewer #3 (Comments to the Authors (Required)):

The authors start by talking about contact sites which sets the stage for the reader to expect more about the role of the NVJ contact site in this phenomenon, and yet no experiments were done to determine whether this is related to the NVJ or can happen in absence of it.

- The reviewer is right that the stage was “too much set” for the reader to expect more experiments regarding the role of the NVJ. The hole formation stems from the nuclear envelope subdomain that surrounds the nucleolus. This same domain is engaged in a tight and dynamic contact with the vacuole, the nucleus-vacuole junction (NVJ). To assess if this participation in two different processes affects hole formation, we have now deleted *NVJ1* and assessed the impact this may have on the formation of nuclear holes. We observe that, in response to zeocin, cells form nuclear holes more readily in the absence of Nvj1 (New Fig 5A, compare fold-change from 0 to 120 min treatment in WT *versus* *nvj1Δ*). Yet, the final hole formation values are unchanged (Fig 5A, time 210 min). We have also performed the experiment in a *yeh2Δ* strain, where the basal level of holes and the readiness to form them upon zeocin treatment is increased. In this case, the absence of Nvj1 does not strengthen the phenotype further (Fig 5A, *yeh2Δ versus yeh2Δ nvj1Δ*). Together, these data suggest that the nuclear envelope subdomain that surrounds the nucleolus is indeed shared between its contact with the vacuole and its availability to engage in nuclear hole formation. As a consequence, we have now adapted our introduction, abstract, and results text in a manner that matches this concept. Thank you.

April 26, 2022

RE: Life Science Alliance Manuscript #LSA-2021-01160-TRR

Dr. María Moriel-Carretero
Centre de Recherche en Biologie cellulaire de Montpellier
(CRBM)
Université de Montpellier,
Centre National de la Recherche Scientifique
1919 Route de Mende
Montpellier 34293
France

Dear Dr. Moriel-Carretero,

Thank you for submitting your revised manuscript entitled "Nuclear ingression of cytoplasmic bodies accompanies a boost in autophagy". We would be happy to publish your paper in Life Science Alliance pending final revisions necessary to meet our formatting guidelines.

- please upload all figure files as individual ones, including the supplementary figure files
- please add your main, supplementary figure, and table legends to the main manuscript text after the references section

FIGURE CHECKS:

- the minimum resolution for all figures should be 300 dpi

A. FINAL FILES:

B. MANUSCRIPT ORGANIZATION AND FORMATTING:

**Submission of a paper that does not conform to Life Science Alliance guidelines will delay the acceptance of your

manuscript.**

The license to publish form must be signed before your manuscript can be sent to production. A link to the electronic license to publish form will be sent to the corresponding author only. Please take a moment to check your funder requirements.

Sincerely,

May 2, 2022

RE: Life Science Alliance Manuscript #LSA-2021-01160-TRRR

Dr. María Moriel-Carretero
Centre de Recherche en Biologie cellulaire de Montpellier (CRBM)
Université de Montpellier,
Centre National de la Recherche Scientifique
1919 Route de Mende
Montpellier 34293
France

Dear Dr. Moriel-Carretero,

Thank you for submitting your Research Article entitled "Nuclear ingression of cytoplasmic bodies accompanies a boost in autophagy". It is a pleasure to let you know that your manuscript is now accepted for publication in Life Science Alliance. Congratulations on this interesting work.

DISTRIBUTION OF MATERIALS:

Again, congratulations on a very nice paper. I hope you found the review process to be constructive and are pleased with how the manuscript was handled editorially. We look forward to future exciting submissions from your lab.

Sincerely,
